# MobileFusion: Mobile-Friendly Infrared and Visible Image Fusion via Structural Re-parameterization

**Yufa Duan** [1]  **Jialing Huang** [1]  **Yingying Wang** [1]  **Weimin Cai** [1]  **Xinghao Ding** [1]  **Xiaotong Tu** [1]

## Abstract

Deep neural networks have recently advanced infrared and visible image fusion (IVIF), but most existing methods rely on sophisticated yet redundant designs, which hinder real-time deployment on mobile devices with limited compute and memory. In this paper, we present MobileFusion, an extremely lightweight and effective convolutional framework that achieves high-quality fusion under strict resource constraints. MobileFusion leverages a novel re-parameterizable multi-branch convolution module to promote cross-modal interactions during training while collapsing into a single-path operator for fast inference. It further incorporates a lightweight attention module to enhance context awareness, together with a re-parameterized feed-forward network to improve feature expressiveness. Extensive experiments demonstrate that MobileFusion delivers a favorable trade-off between fusion quality and computational efficiency, enabling real-time and high-quality IVIF on resource-constrained platforms. The source code is available at `https://github.com/sucessfullys/MobileFusion`.

## 1. Introduction

Image fusion (Liu et al., 2018) aims to integrate complementary information from multiple sensing modalities to produce a more comprehensive representation. Infrared and visible image fusion (IVIF) (Zhao et al., 2023b), as a crucial subfield, has gained increasing attention with broad applications in autonomous driving (Bijelic et al., 2020),

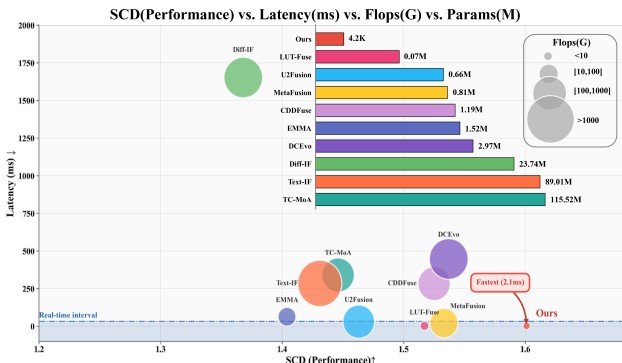

*Figure 1.* MobileFusion delivers outstanding performance while achieving the fastest inference speed, compared with state-of-the-art methods and ultra-fast LUT-Fuse on the FMB dataset.

military reconnaissance (Liu et al., 2022), *etc*. IVIF effectively merges the salient thermal radiation from infrared (IR) images with the rich texture details in visible (VI) images. In recent years, the rapid advancements in deep learning have dramatically accelerated the progress of IVIF technologies (Zheng et al., 2024). In particular, Transformer-based (Ma et al., 2022; Zhao et al., 2023c; Yi et al., 2024b) and diffusion-based (Zhao et al., 2023d; Yi et al., 2024a) models have been designed to improve fusion performance. However, a critical limitation is that these studies predominantly focus on fusion performance metrics at the expense of real-time processing considerations. Their reliance on computationally expensive self-attention mechanisms and iterative diffusion processes prevents them from achieving real-time operational efficiency, even when implemented on the NVIDIA GeForce RTX 4090 platform, as shown in Fig. 1. Recently, LUT-Fuse (Yi et al., 2025) proposes a lookup-table-based ultra-fast IVIF framework that achieves significantly accelerated speed. LUT-Fuse employs a lookup table that integrates low-order approximation encoding and high-level contextual scene encoding, and leverages a distillation strategy to integrate the fusion performance of a pre-trained multi-modal fusion network into the lookup table. However, the representational capacity of LUT-Fuse is inherently limited by the fixed mapping structure of its lookup table, making it difficult to model content-adaptive fusion strategies. Moreover, reliance on a pre-trained teacher net-

[1]Key Laboratory of Multimedia Trusted Perception and Efficient Computing, Ministry of Education of China, Xiamen University, Xiamen, Fujian, China. Correspondence to: Xiaotong Tu <xttu@xmu.edu.cn>.

*Proceedings of the 43rd International Conference on Machine Learning*, Seoul, South Korea. PMLR 306, 2026. Copyright 2026 by the author(s).

work introduces additional design complexity and hinders end-to-end optimization. Therefore, achieving a favorable trade-off between speed and fusion performance under real-time constraints has become a critical research priority.

To address these challenges, we propose MobileFusion, an extremely lightweight and effective method designed for real-time multi-modal image fusion on mobile devices. MobileFusion discards complex architectural designs and instead adopts a streamlined topology with hardware-friendly operators to strike an optimal balance between fusion performance, inference speed, and deployment feasibility. Specifically, we decouple the training and inference phases through structural re-parameterization. During training, we adopt a novel re-parameterizable multi-branch feed-forward structure to facilitate optimization and enhance feature learning. At inference, these branches are analytically fused into an equivalent single-path convolutional layer, thereby simplifying the forward pipeline for efficient deployment and substantially reducing latency and memory overhead, without sacrificing the expressive capacity learned during training. The core design lies in the proposed Re-parameterizable Multi-branch Convolution (RepMBConv) module, which captures multi-scale fusion features through parallel convolutional branches with diverse kernel sizes. These features are first fused via channel-wise concatenation and then passed through a compression-and-projection stage to match the target output dimensionality, yielding a compact representation. Compared to previous Rep-based methods, as illustrated in Appendix B, this design expands the effective receptive field and provides a stronger inductive bias for modality interactions, enabling more effective integration of complementary cross-modal cues.

Building upon the RepMBConv, we further introduce a Lightweight Context-Aware Attention (LCAA) module that jointly captures global semantics and local details. LCAA applies a dual-stream channel attention to recalibrate modality-specific channels, followed by a dual-direction spatial attention that aggregates horizontal and vertical context to generate fine-grained spatial features, effectively highlighting structural regions (e.g., edges and contours) while suppressing redundancy. Moreover, since the linear operations in RepMBConv can limit feature expressiveness, we incorporate a Re-parameterized Feed-Forward Network (RepFFN) to strengthen nonlinearity and improve generalization. With its extremely lightweight and effective design, MobileFusion, with approximately 4K parameters, strikes an optimal balance between real-time efficiency and high-quality fusion within a unified framework. Overall, our contributions of this paper can be summarized as follows:

- We propose MobileFusion, an extremely lightweight and effective framework for real-time infrared and visible image fusion on mobile devices, featuring compact

and efficient modules.

- We propose a Re-parameterizable Multi-branch Convolution (RepMBConv) to encourage cross-modal interactions, along with a Lightweight Context-Aware Attention (LCAA) and a Re-parameterized Feed-Forward Network (RepFFN) to facilitate context-adaptive fusion and enhance nonlinearity.

- MobileFusion achieves a favorable balance between real-time efficiency and fusion quality, highlighting its practical value for real-world IVIF applications on resource-constrained platforms.

## 2. Related Work

### 2.1. Deep Learning-based Image Fusion

Deep learning–based multi-modal image fusion methods have achieved remarkable progress, with a primary focus on improving fusion quality and strengthening semantic representations. In the early stages, pre-trained autoencoders (Jian et al., 2020) served as a dominant paradigm. However, these methods relied on handcrafted fusion rules, which posed significant challenges for end-to-end optimization. Subsequently, end-to-end CNN-based fusion architectures (Ma et al., 2021; Zhao et al., 2023a) were proposed, enabling adaptive fusion and providing greater flexibility and robustness. More recently, Transformer-based (Liu et al., 2023) and diffusion-based (Yi et al., 2024a) methods have introduced novel insights to further improve fusion performance. Despite their impressive results, these methods remain computationally prohibitive for real-time deployment on mobile devices, failing to meet the stringent latency and efficiency requirements of practical applications. Recently, LUT-Fuse (Yi et al., 2025) introduces an innovative framework that combines the lookup table with knowledge distillation, achieving a significant breakthrough in real-time inference speed. However, it still lacks a favorable balance between computational efficiency and fusion performance.

### 2.2. Efficient Architectures Design

Efficient neural networks (Zhang et al., 2023) aim to strike a balance between performance and computational efficiency. Architectures such as MobileNet(Howard et al., 2019) and ShuffleNet (Zhang et al., 2018) leveraged depthwise separable convolutions and channel shuffling to achieve high performance with drastically reduced FLOPs, while StarNet (Ma et al., 2024) and FasterNet (Chen et al., 2023) generated expressive feature maps through cost-effective operations. Furthermore, structural re-parameterization has emerged as a powerful paradigm in efficient network design. Methods such as RepVGG (Ding et al., 2021) MobileOne (Vasu et al., 2023b), and RepViT (Wang et al., 2024a) employed re-

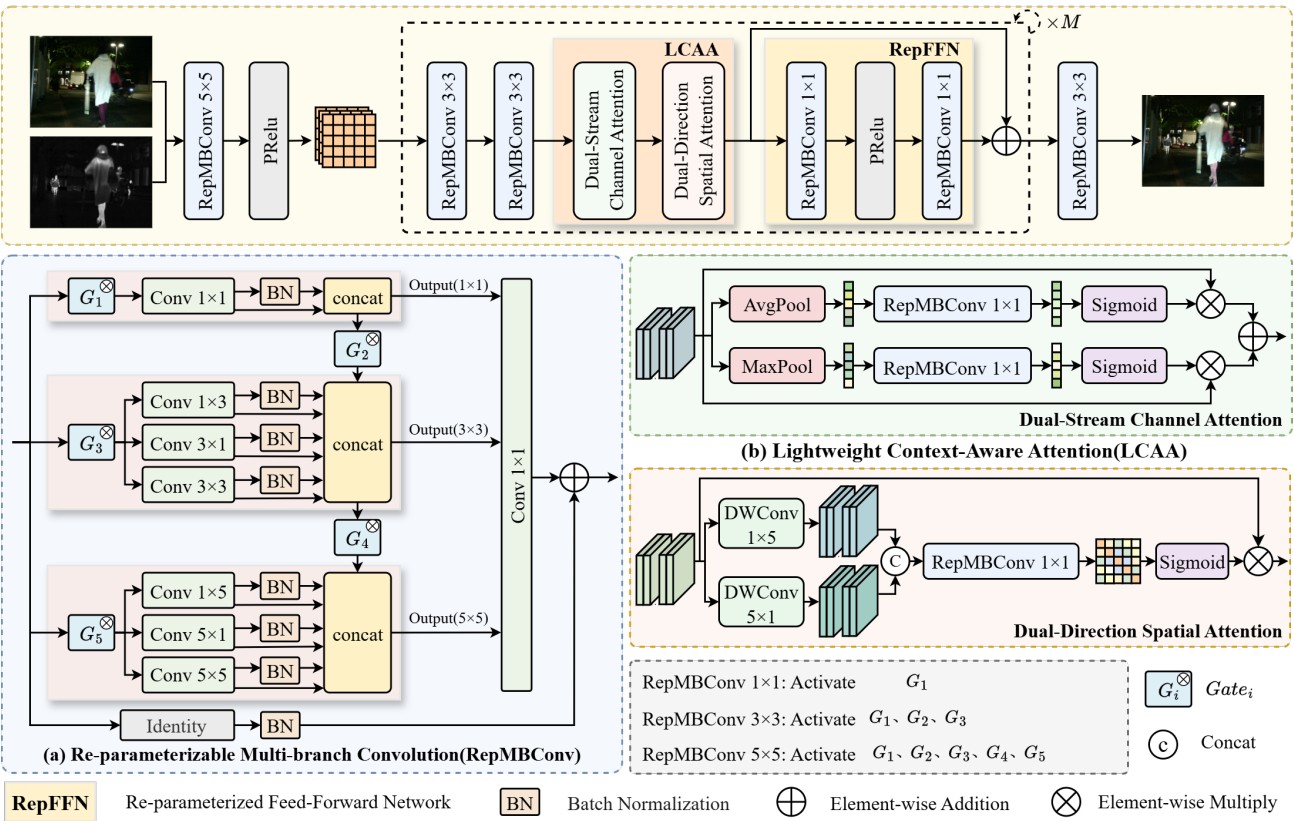

*Figure 2.* The overall workflow of MobileFusion. MobileFusion consists of three core components: Re-parameterizable Multi-Branch Convolution (RepMBConv), Lightweight Context-Aware Attention (LCAA), and Re-parameterized Feed-Forward Network (RepFFN). During inference, RepMBConv is re-parameterized into a standard convolution for fast inference while preserving the training performance.

parameterization techniques within their core architectures. This strategy preserves rich representational capacity during training while enabling hardware-friendly, low-latency execution at inference. However, structural re-parameterization is primarily designed for high-level vision tasks, focusing on overall semantic recognition, and lacks a modality interaction mechanism, making its direct application to low-level IVIF unsatisfactory (Lu et al., 2023). This motivates the design of a structured re-parameterization fusion architecture that delivers both real-time efficiency on mobile devices and high-quality fusion results.

## 3. Methodology

In this section, we initially present the comprehensive workflow of our proposed methodology. Subsequently, we provide a detailed exposition of the proposed Re-parameterizable Multi-branch Convolution (RepMBConv) module, Lightweight Context-Aware Attention (LCAA) module, and Re-parameterized Feed-Forward Network (RepFFN). The section concludes with a thorough specification of loss functions.

### 3.1. Overall Workflow

Given an input pair of visible and infrared images $\{I_{vi} \in \mathbb{R}^{H \times W \times 3}, \ I_{ir} \in \mathbb{R}^{H \times W \times 1}\}$, where H and W denote the height and width of the images, respectively, we design a re-parameterizable multi-branch convolutional network to generate a fused image $I_f \in \mathbb{R}^{H \times W \times 3}$. This process is mathematically formulated as:

$$I_f = \mathcal{N}\left(\{I_{vi}, \ I_{ir}\}; \theta_{\mathcal{N}}\right), \tag{1}$$

where $\mathcal{N}$ denotes the training-time multi-branch fusion network and $\theta_{\mathcal{N}}$ represents its learnable parameters. During inference, we re-parameterize $\mathcal{N}$ into an equivalent single-path network $\mathcal{N}_{\text{rep}}$. Specifically, we employ a deterministic mapping $\mathcal{M}(\cdot)$ that analytically merges all branches and their associated normalization parameters into a compact set of convolutional weights. The process can be defined as:

$$\theta_{\text{rep}} = \mathcal{M}(\theta_{\mathcal{N}}), \quad I_f = \mathcal{N}_{\text{rep}}\left(\{I_{\text{vi}}, \ I_{\text{ir}}\}; \theta_{\text{rep}}\right). \tag{2}$$

The re-parameterized network $\mathcal{N}_{\text{rep}}$ is functionally equivalent to the training-time network $\mathcal{N}$, while reducing memory-access overhead and inference latency. Based on this structural re-parameterization principle, we develop MobileFusion, whose overall pipeline is shown in Fig. 2.

Specifically, during training, the source images are first concatenated and fed into a RepMBConv layer with a $5 \times 5$ kernel followed by PReLU to produce an initial fusion representation. The shallow representation is then passed through a stack of $M$ lightweight blocks to progressively extract deeper fusion features, which are finally reconstructed into the fused image via a RepMBConv layer with a $3 \times 3$ kernel. During inference, all RepMBConv layers are analytically re-parameterized into standard convolutions, yielding a compact single-branch network. This streamlined design is hardware-friendly and deployable on mobile devices.

### 3.2. Re-parameterizable Multi-Branch Convolution

Structural re-parameterization has proven effective for high-level vision tasks, yet it degrades performance when naively applied to multi-modal fusion, since conventional re-parameterized designs do not effectively encourage cross-modal interactions. To address this limitation, we introduce RepMBConv (Fig. 2(a)), a re-parameterizable multi-branch convolution module tailored for real-time IVIF.

#### 3.2.1. TRAINING-TIME MULTI-BRANCH

In contrast to previous re-parameterized designs that mainly rely on a single $k \times k$ path (e.g., $k = 3$), RepMBConv employs a wider and more heterogeneous set of branches, combining standard convolutions and convolutions with BatchNorm while additionally incorporating heterogeneous kernel geometries ($1 \times k$ and $k \times 1$). The multi-branch outputs are first aggregated by channel-wise concatenation and then projected through a subsequent $1 \times 1$ convolution to obtain the fused representation. In parallel, we employ an identity skip branch followed by BatchNorm to facilitate optimization and stabilize training. The skip branch is added to the fused main path to produce the final output. This design expands the effective receptive field, facilitates richer multi-modal features extraction, and provides a strong inductive bias for modality interactions.

#### 3.2.2. DEPLOYMENT-TIME RE-PARAMETERIZATION

At inference, all branches are analytically merged into a single standard $k \times k$ convolution, resulting in a single-path and mobile-friendly operator.

**(i) Folding BatchNorm into convolution.** For the $i$-th branch, we denote the convolution parameters as ($W_i' \in \mathbb{R}^{c_i \times c_{\text{in}} \times k_i^h \times k_i^w}, b_i' \in \mathbb{R}^{c_i}$). If the branch is followed by a BatchNorm with parameters ($\gamma_i, \beta_i, \mu_i, \sigma_i^2$), we fold BatchNorm into an equivalent convolution:

$$W_i = \frac{\gamma_i}{\sqrt{\sigma_i^2 + \epsilon}} W_i', \quad b_i = \frac{\gamma_i}{\sqrt{\sigma_i^2 + \epsilon}}(b_i' - \mu_i) + \beta_i, \quad (3)$$

where $c_{\text{in}}$ denotes the number of input channels, $c_i$ is the number of output channels of branch $i$, ($k_i^h, k_i^w$) is its kernel

size. $\epsilon > 0$ is a small constant for numerical stability. Branches without BatchNorm keep ($W_i, b_i$) = ($W_i', b_i'$).

**(ii) Aligning kernels and aggregating branches.** To enable fusion across heterogeneous kernel geometries, we zero-pad each branch $W_i$ to the common spatial size $k \times k$, yielding $\widehat{W}_i \in \mathbb{R}^{c_i \times c_{\text{in}} \times k \times k}$. We then aggregate all padded kernels and biases along the output-channel dimension. Let $\phi(\cdot)$ reshape a kernel tensor in $\mathbb{R}^{c_i \times c_{\text{in}} \times k \times k}$ into a matrix in $\mathbb{R}^{c_i \times (c_{\text{in}} k^2)}$ by flattening the last three dimensions. The process is formulated as:

$$U = \left[\phi(\widehat{W}_1)^\top, \phi(\widehat{W}_2)^\top, \ldots, \phi(\widehat{W}_N)^\top\right]^\top \in \mathbb{R}^{C \times (c_{\text{in}} k^2)}, \quad (4)$$

$$\widetilde{W} = \phi^{-1}(U) \in \mathbb{R}^{C \times c_{\text{in}} \times k \times k}, \widetilde{b} = \left[b_1^\top, \ldots, b_N^\top\right]^\top \in \mathbb{R}^C, \quad (5)$$

where $N$ is the number of branches and $C = \sum_{i=1}^N c_i$.

**(iii) Fusing the subsequent $1 \times 1$ projection.** We parameterize the subsequent $1 \times 1$ convolution by weight $P \in \mathbb{R}^{c_{\text{out}} \times C \times 1 \times 1}$ and bias $b_P \in \mathbb{R}^{c_{\text{out}}}$. Since a $1 \times 1$ convolution performs a channel-wise linear projection, it linearly combines the $C$ intermediate channels produced by the aggregated branches. Consequently, the multi-branch main path can be re-parameterized into a single $k \times k$ convolution ($W_{\text{main}} \in \mathbb{R}^{c_{\text{out}} \times c_{\text{in}} \times k \times k}, b_{\text{main}} \in \mathbb{R}^{c_{\text{out}}}$):

$$W_{\text{main}}[o, :, :, :] = \sum_{j=1}^C P[o, j] \, \widetilde{W}[j, :, :, :], \quad (6)$$

$$b_{\text{main}}[o] = \sum_{j=1}^C P[o, j] \, \widetilde{b}[j] + b_P[o], \quad (7)$$

where $o \in \{1, \ldots, c_{\text{out}}\}$ and $c_{\text{out}}$ denotes the number of output channels of RepMBConv.

**(iv) Merging the identity skip branch.** When the number of input and output channels are equal ($c_{\text{in}} = c_{\text{out}}$), we introduce an identity skip branch. Finally, we integrate the identity skip branch with BatchNorm into the main path. We represent the identity mapping as a $k \times k$ convolution with a Dirac kernel $W_s'$ and zero bias $b_s' = 0$. The Dirac kernel $W_s' \in \mathbb{R}^{c_{\text{out}} \times c_{\text{in}} \times k \times k}$ is defined as:

$$[W_s']_{c_{\text{out}}, c_{\text{in}}, u, v} = \begin{cases} 1, & (u, v) = \left(\lfloor \frac{k}{2} \rfloor, \lfloor \frac{k}{2} \rfloor\right), \\ 0, & \text{otherwise.} \end{cases} \quad (8)$$

We fold the BatchNorm parameters of the skip branch using Eq. (3) to obtain an equivalent convolution ($W_s, b_s$). The final deployed single-path $k \times k$ convolution parameters ($W_{\text{rep}}, b_{\text{rep}}$) are:

$$W_{\text{rep}} = W_{\text{main}} + W_s, \quad b_{\text{rep}} = b_{\text{main}} + b_s. \quad (9)$$

### 3.3. Lightweight Context-Aware Attention

To improve context-aware fusion, we propose a LCAA module, as shown in Fig. 2(b). LCAA jointly models channel

and spatial dependencies to adaptively capture global semantics and local details. In the dual-stream channel attention, we employ both adaptive average pooling and max pooling to summarize global statistics into channel descriptors, dynamically modulating the importance of each channel. We additionally introduce max pooling allowing for the capture of significant features in infrared images. The entire process can be expressed as:

$$Z_{\text{avg}} = Z_c^{\text{avg}} = \frac{1}{HW} \sum_{i=1}^{H} \sum_{j=1}^{W} F_{i,j,c},$$

$$Z_{\text{max}} = Z_c^{\text{max}} = \max_{1 \leq i \leq H, \, 1 \leq j \leq W} F_{i,j,c},$$

(10)

$$A_{\text{avg}} = \sigma\big(\text{RepMBConv}_{1\times1}(Z_{\text{avg}})\big),$$

$$A_{\text{max}} = \sigma\big(\text{RepMBConv}_{1\times1}(Z_{\text{max}})\big),$$

(11)

$$\hat{F} = (A_{\text{avg}} * F) + (A_{\text{max}} * F),$$

(12)

where $F$ denotes the input features, and $F_{i,j,c}$ is the feature value at spatial location $(i,j)$ in channel $c$. $\sigma(\cdot)$ denotes the sigmoid activation. $A_{\text{avg}}$ and $A_{\text{max}}$ represent the global attention weights. $\hat{F}$ is the recalibrated features. To further highlight fine-grained local structures, we introduce a dual-direction spatial attention. It models anisotropic spatial context by applying depthwise convolutions independently along the horizontal and vertical directions. The resulting directional features are concatenated and projected with a $1 \times 1$ RepMBConv followed by a sigmoid gate, yielding a spatial weighting map that adaptively reweights $\hat{F}$ to emphasize critical local spatial details. It can be expressed as:

$$\hat{F}_w = \text{DWConv}_{1\times5}(\hat{F}),$$

$$\hat{F}_h = \text{DWConv}_{5\times1}(\hat{F}),$$

(13)

$$A_s = \sigma\Big(\text{RepMBConv}_{1\times1}\big(\text{Cat}\big[\hat{F}_w, \, \hat{F}_h\big]\big)\Big),$$

(14)

$$\bar{F} = A_s * \hat{F},$$

(15)

where $\text{DWConv}_{1\times5}(\cdot)$ and $\text{DWConv}_{5\times1}(\cdot)$ denote depthwise convolutions with kernel sizes $1 \times 5$ and $5 \times 1$, respectively. $\text{Cat}[\cdot]$ is channel-wise concatenation. $A_s$ represents spatial attention weights. $\bar{F}$ denotes the refined features.

### 3.4. Re-parameterized Feed-Forward Network

While RepMBConv captures multi-scale features efficiently and promotes cross-modal interactions, it relies primarily on linear convolutional mixing, which limits feature transformations. To address this, we introduce RepFFN, a convolutional feed-forward module built from $1 \times 1$ RepMBConvs with nonlinear activation, boosting nonlinear expressiveness as well as generalization. Specifically, it adopts a two-stage $1 \times 1$ RepMBConv architecture with a channel expansion ratio of $r = 2$ and a PReLU activation in between to increase

feature diversity. At inference time, the multi-branch $1 \times 1$ RepMBConvs are analytically re-parameterized into a single convolution, yielding efficient nonlinear enhancement with minimal overhead. This process can be formulated as:

$$D = \rho\big(\text{RepMBConv}_{1\times1}(\bar{F})\big),$$

$$\text{RepFFN}(\bar{F}) = \bar{F} + \text{RepMBConv}_{1\times1}(D),$$

(16)

where $D \in \mathbb{R}^{H \times W \times rC}$ denotes the intermediate features, and $\rho(\cdot)$ denotes the PReLU activation.

### 3.5. Loss Functions

In the training process, we optimize a weighted combination of fusion-specific objectives, including structural similarity (SSIM) loss (Zhao et al., 2016), intensity loss, maximum gradient loss, and color consistency loss. The SSIM loss enables structural similarity between the fused images $I_f$ and the source images $\{I_{vi}, \, I_{ir}\}$. It is defined as:

$$\mathcal{L}_{\text{ssim}} = (1 - \text{SSIM}(I_f, I_{vi})) + (1 - \text{SSIM}(I_f, I_{ir})). \quad (17)$$

The intensity loss aligns the fused image with the element-wise maximum of the infrared and visible inputs, encouraging the preservation of the strongest intensity responses from both modalities and thereby retaining salient targets. This is defined as:

$$\mathcal{L}_{\text{int}} = \frac{1}{HW} \left\| I_f - \max(I_{vi}, I_{ir}) \right\|_1, \quad (18)$$

where $\max(\cdot)$ denotes the element-wise maximum selection. $\| \cdot \|_1$ is the $l_1$-norm. The maximum-gradient loss preserves sharp edges and fine textures by aligning the gradients of the fused image with the element-wise maximum of the gradients from the infrared and visible inputs. It can be expressed as:

$$\mathcal{L}_{\text{grad}} = \frac{1}{HW} \left\| \nabla I_f - \max(\nabla I_{vi}, \nabla I_{ir}) \right\|_1, \quad (19)$$

where $\nabla$ represents the Sobel operator. The color consistency loss is introduced to preserve chromatic fidelity from the visible source images. It is defined as:

$$\mathcal{L}_{\text{color}} = \frac{1}{HW} \left\| \Phi_{CbCr}(I_f) - \Phi_{CbCr}(I_{vi}) \right\|_1, \quad (20)$$

where $\Phi_{CbCr}$ represents the transformation from RGB to CbCr color space. Finally, the overall loss is defined as:

$$\mathcal{L}_{total} = \lambda_1 \cdot \mathcal{L}_{\text{ssim}} + \lambda_2 \cdot \mathcal{L}_{\text{int}} + \lambda_3 \cdot \mathcal{L}_{\text{grad}} + \lambda_4 \cdot \mathcal{L}_{\text{color}}, \quad (21)$$

where $\lambda_1$, $\lambda_2$, $\lambda_3$ and $\lambda_4$ are hyper-parameters.

## 4. Experiments

### 4.1. Experimental Setup

#### 4.1.1. IMPLEMENTATION DETAILS

We train MobileFusion for 120 epochs using the AdamW optimizer with a learning rate of $1 \times 10^{-5}$, implemented

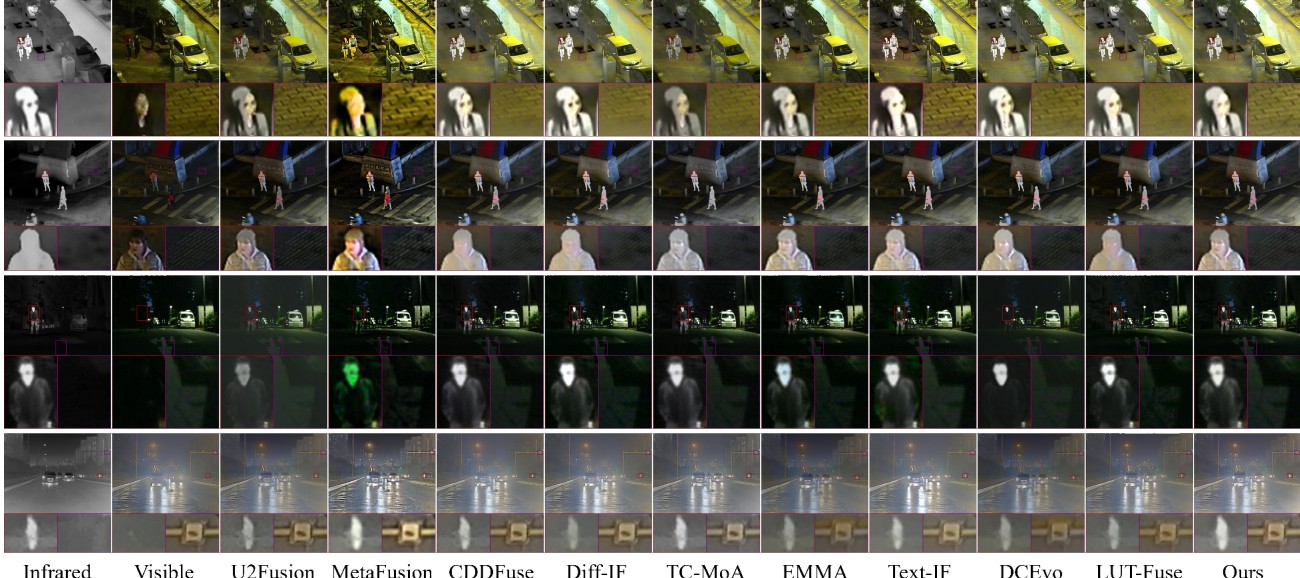

Infrared    Visible    U2Fusion    MetaFusion    CDDFuse    Diff-IF    TC-MoA    EMMA    Text-IF    DCEvo    LUT-Fuse    Ours

*Figure 3.* Qualitative comparison of MobileFusion with the state-of-the-art multi-modal image fusion methods. From top to bottom: two groups of data from LLVIP, data from MSRS, and data from FMB datasets, respectively. **Please zoom in for better viewing**.

*Table 1.* Quantitative comparison of MobileFusion with state-of-the-art multi-modal image fusion methods. ↑: higher is better; ↓: lower is better. (Highest value is marked in Red and the second-highest in Blue).

| Methods | MSRS Dataset | | | | | FMB Dataset | | | | | LLVIP Dataset | | | | |
|---|---|---|---|---|---|---|---|---|---|---|---|---|---|---|---|
| | EN↑ | SCD↑ | CC↑ | SSIM↑ | MI↑ | EN↑ | SCD↑ | CC↑ | SSIM↑ | MI↑ | EN↑ | SCD↑ | CC↑ | SSIM↑ | MI↑ |
| U2Fusion (TPAMI'22) | 6.242 | 1.318 | 0.610 | 0.860 | 2.799 | 6.658 | 1.463 | 0.618 | 0.965 | 3.153 | 6.851 | 1.406 | 0.710 | 0.846 | 2.681 |
| MetaFusion (CVPR'23) | 6.378 | 1.499 | 0.606 | 0.778 | 1.663 | 6.258 | 1.533 | 0.610 | 0.872 | 2.503 | 6.918 | 1.322 | 0.683 | 0.732 | 1.748 |
| CDDFuse (CVPR'23) | 6.685 | 1.682 | 0.598 | 0.936 | 3.124 | 6.352 | 1.525 | 0.614 | 0.957 | 3.122 | 7.157 | 1.516 | 0.713 | 0.924 | 2.725 |
| Diff-IF (Image Fusion'24) | 6.581 | 1.624 | 0.602 | 0.944 | 3.145 | 6.622 | 1.368 | 0.583 | 0.963 | 3.255 | 7.217 | 1.433 | 0.669 | 0.922 | 2.919 |
| TC-MoA (CVPR'24) | 6.616 | 1.685 | 0.609 | 0.845 | 2.272 | 6.647 | 1.446 | 0.595 | 0.969 | 2.368 | 7.300 | 1.502 | 0.674 | 0.933 | 2.244 |
| EMMA (CVPR'24) | 6.653 | 1.629 | 0.596 | 0.913 | 2.940 | 6.673 | 1.404 | 0.605 | 0.955 | 2.795 | 7.211 | 1.493 | 0.679 | 0.918 | 2.291 |
| Text-IF (CVPR'24) | 6.667 | 1.689 | 0.607 | 0.937 | 3.274 | 6.672 | 1.431 | 0.589 | 0.970 | 3.287 | 7.326 | 1.520 | 0.680 | 0.921 | 2.803 |
| DCEvo (CVPR'25) | 6.622 | 1.661 | 0.604 | 0.964 | 2.881 | 6.701 | 1.537 | 0.604 | 0.972 | 3.127 | 7.308 | 1.561 | 0.672 | 0.940 | 2.806 |
| LUT-Fuse (ICCV'25) | 6.628 | 1.627 | 0.612 | 0.932 | 2.513 | 6.618 | 1.517 | 0.606 | 0.976 | 3.382 | 7.202 | 1.526 | 0.674 | 0.914 | 2.928 |
| **MobileFusion (ours)** | 6.670 | 1.766 | 0.616 | 0.956 | 3.172 | 6.685 | 1.601 | 0.623 | 0.986 | 3.299 | 7.335 | 1.653 | 0.721 | 0.935 | 2.922 |

on the NVIDIA GeForce RTX 4090 GPU with PyTorch framework. The batch size is set to 12. During training, source images are randomly cropped to $96 \times 96$. The set of hyper-parameters is $\lambda_1 = 1$, $\lambda_2 = 2$, $\lambda_3 = 10$, and $\lambda_4 = 12$. We evaluated performance on the GeForce RTX 4090 GPU and NVIDIA Jetson Orin NX edge platform to assess mobile deployment feasibility. More architectural details are provided in Appendix A.

### 4.1.2. DATASETS

To evaluate the effectiveness of our proposed MobileFusion, we conduct extensive experiments on three public infrared and visible fusion benchmarks, including MSRS (Tang et al., 2022), FMB (Liu et al., 2023), and LLVIP (Jia et al., 2021). The image resolutions of MSRS, FMB, and LLVIP are 640×480, 800×600, and 1280×1024, respectively. We use a total of 1500 image pairs for training, while 100 image pairs from MSRS, 100 from FMB, and 100 from

LLVIP are utilized for testing.

### 4.1.3. BASELINES & METRICS

We compare our proposed MobileFusion with nine state-of-the-art methods on multiple datasets, including U2Fusion(Xu et al., 2020), MetaFusion(Zhao et al., 2023a), CDDFuse(Zhao et al., 2023c), Diff-IF(Yi et al., 2024a), TC-MoA(Zhu et al., 2024), Text-IF(Yi et al., 2024b), EMMA(Zheng et al., 2024), LUT-Fuse(Yi et al., 2025), and DCEvo(Liu et al., 2025). For quantitative comparisons, we employ the metrics including mutual information (MI) (Qu et al., 2002), information entropy (EN) (Roberts et al., 2008), correlation coefficient (CC), structural similarity index measure (SSIM) (Wang et al., 2004), and sum of the correlations of differences (SCD) (Aslantas & Bendes, 2015). Higher values of EN, CC, MI, SSIM, and SCD indicate higher quality of the fusion image.

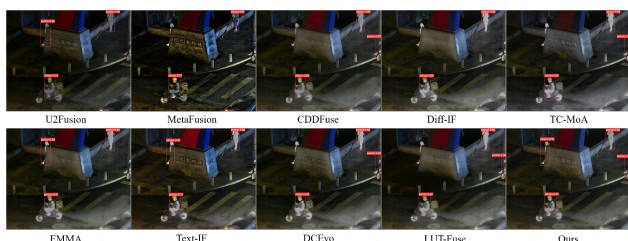

*Figure 4.* Qualitative comparison in object detection task on LLVIP.

## 4.2. Qualitative Experiments

We present a qualitative comparison of various methods in Fig. 3, where three key advantages of our method are clearly shown. First, the significant characteristics of infrared images are effectively integrated and highlighted. For instance, U2Fusion, MetaFusion, TC-MoA, and EMMA struggle with preserving thermal information. In contrast, our method excels at retaining infrared features, particularly in complex scenes where thermal cues are indispensable for accurate scene understanding. Second, our method exhibits cleaner structures and sharper textures in fused results, with greater robustness in noisy low-light environments. By comparison, LUT-Fuse, Diff-IF, DCEvo, and CDDFuse tend to lose fine-grained details, resulting in noticeable blur and texture degradation (e.g., cobblestone roads and road signs). Furthermore, unlike MetaFusion and Text-IF, our method maintains consistent chromaticity and produces visually more natural and pleasing results. This indicates that our fusion process preserves structural and intensity information without sacrificing color fidelity. Overall, our method achieves high-quality fusion with a lightweight architecture, making our method attractive for practical deployment under limited computational budgets.

## 4.3. Quantitative Experiments

The quantitative results on multiple benchmarks are summarized in Tab. 1. Note that MobileFusion achieves the best overall performance across MSRS, FMB, and LLVIP datasets on all reported metrics. In particular, higher SCD and EN scores indicate that our method preserves more salient information from the source modalities. Meanwhile, the gains in CC, MI, and SSIM indicate that our method achieves improved structural consistency and fidelity to the inputs, with richer edge details and reduced distortion. Compared to LUT-Fuse, MobileFusion achieves superior performance with a smaller model size and faster inference.

## 4.4. Performance on High-Level Tasks

To further validate the effectiveness of our fusion results in downstream high-level vision tasks, we conduct object detection and semantic segmentation experiments on the fused

*Table 2.* Quantitative comparison of detection performance.

| Methods | Precision | Recall | mAP@0.5 | mAP@0.5:0.95 |
|---|---|---|---|---|
| U2Fusion | 0.9307 | **0.8874** | 0.9371 | 0.5836 |
| MetaFusion | 0.9220 | 0.8396 | 0.9160 | 0.5384 |
| CDDFuse | 0.9436 | 0.8609 | 0.9365 | **0.5918** |
| Diff-IF | 0.9319 | 0.8529 | 0.9296 | 0.5725 |
| TC-MoA | 0.9183 | 0.8283 | 0.9010 | 0.5293 |
| EMMA | 0.9417 | 0.8636 | 0.9343 | 0.5778 |
| Text-IF | 0.9327 | 0.8518 | 0.9360 | 0.5690 |
| DCEvo | 0.9443 | 0.8422 | 0.9295 | 0.5722 |
| LUT-Fuse | 0.9366 | 0.8369 | 0.9289 | 0.5692 |
| **MobileFusion** | **0.9585** | 0.8643 | **0.9382** | 0.5884 |

*Table 3.* Efficiency comparisons on NVIDIA GeForce RTX 4090. (**Bold:** the best)

| Methods | Params/M | FLOPs/G | Latency/ms | Real-Time |
|---|---|---|---|---|
| U2Fusion | 0.66 | 641.52 | 30.60 | ✓ |
| MetaFusion | 0.81 | 394.54 | 21.04 | ✓ |
| CDDFuse | 1.19 | 687.26 | 283.52 | ✗ |
| Diff-IF | 23.74 | 1419.45 | 1652.99 | ✗ |
| TC-MoA | 115.52 | 710.20 | 340.60 | ✗ |
| EMMA | 1.52 | 65.77 | 43.74 | ✗ |
| Text-IF | 89.01 | 2403.76 | 285.35 | ✗ |
| DCEvo | 2.97 | 1445.05 | 447.22 | ✗ |
| LUT-Fuse | 0.07 | 3.84 | 3.41 | ✓ |
| **MobileFusion** | **0.004** | **1.77** | **2.10** | ✓ |

images. We adopt YOLOv5[1] as the object detection backbone and train it on the source images of LLVIP. Qualitative and quantitative experimental results are presented in Fig. 4 and Tab. 2. In terms of qualitative comparison, our method successfully detects all objects in the scene, whereas other approaches miss some detections. In terms of the quantitative comparison, our method achieves the best detection performance, demonstrating its robust representations for downstream detection. The semantic segmentation results are provided in Appendix C.

## 4.5. Computational Efficiency Comparison

Real-time performance is crucial for practical deployment of image fusion in real-world applications such as autonomous driving and surveillance. To this end, we evaluate the computational efficiency of various methods on both desktop (PC) and mobile platforms. We report the average value of ten latency results, where an inference time of less than 33 ms is regarded as real-time performance.

### 4.5.1. PC PLATFORM

We present the computational efficiency comparison on an NVIDIA GeForce RTX 4090 platform. As reported in Tab. 3, even on the high-performance computing platform with

---

[1] https://github.com/ultralytics/yolov5

*Table 4.* Efficiency comparison on NVIDIA Jetson Orin NX.

| Methods | Params/M | FLOPs/G | Latency/ms | Real-Time |
|---|---|---|---|---|
| U2Fusion | 0.66 | 405.17 | 246.67 | × |
| MetaFusion | 0.81 | 249.38 | 480.46 | × |
| CDDFuse | 1.19 | 547.62 | 2652.97 | × |
| Diff-IF | 23.74 | 757.04 | 8024.35 | × |
| TC-MoA | 115.52 | 455.13 | 3552.48 | × |
| EMMA | 1.52 | 41.54 | 451.37 | × |
| Text-IF | 89.01 | 1518.88 | 2782.75 | × |
| DCEvo | 2.97 | 770.67 | 5740.24 | × |
| LUT-Fuse | 0.07 | 2.42 | 17.41 | ✓ |
| **Mobile-Fusion** | **0.004** | **1.13** | **12.88** | ✓ |

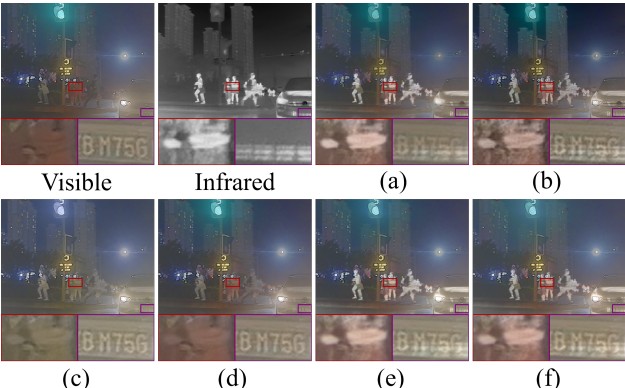

| Visible | Infrared | (a) | (b) |
|---|---|---|---|
| (c) | (d) | (e) | (f) |

*Figure 5.* Qualitative results of the ablation experiments.

*Table 5.* Quantitative results of the ablation experiments.

| RepMBConv | LCAA | RepFFN | EN | SCD | CC | SSIM | MI |
|---|---|---|---|---|---|---|---|
| ✓ | ✓ | | 6.672 | 1.584 | 0.616 | 0.975 | 3.291 |
| ✓ | | ✓ | 6.614 | 1.529 | 0.604 | 0.958 | 3.287 |
| ✓ | ✓ | ✓ | **6.685** | **1.601** | **0.623** | **0.986** | **3.299** |

*Table 6.* Quantitative results of the ablation experiments.

| Method | EN↑ | SCD↑ | CC↑ | SSIM↑ | MI↑ |
|---|---|---|---|---|---|
| Conv | 6.228 | 1.320 | 0.487 | 0.820 | 2.489 |
| Rep | 6.474 | 1.431 | 0.572 | 0.916 | 2.973 |
| RepMBConv (inference) | 6.685 | 1.601 | 0.623 | 0.986 | 3.299 |
| RepMBConv (training) | **6.693** | **1.608** | **0.631** | **0.991** | **3.305** |

a power consumption of 450W, most existing methods fail to achieve real-time performance, severely limiting their practical applicability in real-world scenarios.

### 4.5.2. MOBILE PLATFORM

We further evaluate computational efficiency on the NVIDIA Jetson Orin NX, a representative edge platform. As reported in Tab. 4, benefiting from the lightweight and mobile-friendly compact architecture, only our method and LUT-Fuse achieve real-time inference with 480P ($640 \times 480$) input images. Compared with LUT-Fuse, our method strikes a favorable balance between inference speed and fusion quality, maintaining real-time throughput while consistently achieving higher fusion quality in both qualitative and quantitative evaluations. These results demonstrate the practicality of our approach for deploying real-time fusion on resource-constrained edge devices.

### 4.6. Ablation Experiments

We conduct ablation studies on the FMB dataset to validate the effectiveness of the proposed components, including the Re-parameterizable Multi-Branch Convolution (RepMBConv) module, the Lightweight Context-Aware Attention (LCAA) module, and the Re-parameterized Feed-Forward

Network (RepFFN). As shown in Fig. 5(a,b), removing LCAA leads to noticeable loss of fine-grained details, while removing RepFFN weakens nonlinear feature transformations, resulting in less expressive representations and degraded fusion performance. These observations highlight the essential roles of both components in the overall architecture. Furthermore, as shown in Fig. 5(c,d), replacing RepMBConv with either standard convolutions or conventional re-parameterized design fails to sufficiently achieve cross-modal interactions, leading to incomplete information aggregation and inferior fusion results. In contrast, RepMBConv expands the effective receptive field, enables richer multi-modal feature extraction, and introduces a strong inductive bias that explicitly encourages modality interactions, thereby producing higher-quality fused images, as shown in Fig. 5(e) (training) and Fig. 5(f) (inference). Moreover, our full model achieves the best fusion metrics, with re-parameterization maintaining the training-time performance, as clearly demonstrated in Tabs. 5 and 6.

## 5. Conclusion

In this paper, we propose MobileFusion, an extremely lightweight and effective convolutional framework for real-time infrared and visible image fusion on resource-constrained mobile devices. MobileFusion adopts re-parameterizable multi-branch convolutions to facilitate representation learning and enhance cross-modal interactions during training, and collapses them into a compact single-path network for fast inference. Combined with a lightweight context-aware attention design and an efficient feed-forward enhancement, MobileFusion achieves a favorable balance between fusion quality and computational efficiency, with approximately 4K parameters. Extensive experiments on multiple benchmarks validate its consistent effectiveness and speed.

## Acknowledgements

This work was supported by the Natural Science Foundation of Xiamen, China under Grant No. 3502Z202573011, and the Fundamental Research Funds for the Central Universities under Grant No. 20720252020.

## Impact Statement

This paper presents work whose goal is to advance the field of Machine Learning. There are many potential societal consequences of our work, none which we feel must be specifically highlighted here.

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

# A. Network Architecture

*Table 7.* MobileFusion detailed architecture specifications.

| Description | Training Stage | | Inference Stage |
|---|---|---|---|
| | RepMBConv5 × 5 (Input=4, output=8) | | |
| | Name | Details | |
| Shallow Features Fusion | Conv5 × 5 | C(4, 48, 5, 1, 2) | Conv5 × 5, C(4, 8, 5, 1, 2) |
| | Conv5 × 1 | C(4, 48, 5, 1, (2, 0)) | |
| | Conv1 × 5 | C(4, 48, 5, 1, (0, 2)) | |
| | Conv3 × 3 | C(4, 48, 3, 1, 1) | |
| | Conv3 × 1 | C(4, 48, 3, 1, (1, 0)) | |
| | Conv1 × 3 | C(4, 48, 3, 1, (0, 1)) | |
| | Conv1 × 1 | C(4, 48, 1, 1, 0) | |
| | Channel-Wise Concatenation | | |
| | Conv1 × 1 | C(672, 8, 1, 1, 0) | |
| | PReLU(Channel = 8) | | PReLU(8) |
| | RepMBConv3 × 3 (Input=8, output=8) | | |
| | Name | Details | |
| Two-Deep Features Extraction | Conv3 × 3 | C(8, 48, 3, 1, 1) | Conv3 × 3, C(8, 8, 3, 1, 1) |
| | Conv3 × 1 | C(8, 48, 3, 1, (1, 0)) | |
| | Conv1 × 3 | C(8, 48, 3, 1, (0, 1)) | |
| | Conv1 × 1 | C(8, 48, 1, 1, 0) | |
| | Channel-Wise Concatenation | | |
| | Conv1 × 1 | C(384, 8, 1, 1, 0) | |
| | RepMBConv3 × 3 (Input=8, output=8) | | Conv3 × 3, C(8, 8, 3, 1, 1) |
| | Lightweight Context-Aware Attention (Input=8, output=8) | | |
| | AdaptiveAvgPool(output size=1) | | AdaptiveAvgPool(1) |
| | RepMBConv1 × 1 (Input=8, output=8) | | Conv1 × 1, C(8, 8, 1, 1, 0) |
| | Name | Details | |
| | Conv1 × 1 | C(8, 48, 1, 1, 0) | |
| | Channel-Wise Concatenation | | |
| | Conv1 × 1 | C(96, 8, 1, 1, 0) | |
| | Sigmoid | | Sigmoid |
| | MaxPool(output size=1) | | MaxPool(1) |
| | RepMBConv1 × 1 (Input=1, output=8) | | Conv1 × 1, C(1, 8, 1, 0, 1) |
| | Sigmoid | | Sigmoid |
| | DWConv 1 × 5 (groups=8) | C(8, 8, (1,5), 1, (0,2)) | DWConv 1 × 5, C(8, 8, (1,5), 1, (0,2)) DWConv 5 × 1, C(8, 8, (5,1), 1, (2,0)) Conv 1 × 1, C(16, 1, 1, 1, 0) |
| | DWConv 5 × 1 (groups=8) | C(8, 8, (5,1), 1, (2,0)) | |
| | Channel-Wise Concatenation | | |
| | RepMBConv1 × 1 (Input=16, output=1) | | |
| | Sigmoid | | Sigmoid |
| | RepMBConv1 × 1 (Input=8, output=16) | | Conv 1 × 1, C(8, 16, 1, 1, 0) |
| | PReLU(Channel = 16) | | PReLU(16) |
| | RepMBConv1 × 1 (Input=16, output=8) | | Conv 1 × 1, C(16, 8, 1, 1, 0) |
| Reconstruction | RepMBConv3 × 3, (Input=8, output=3) | | Conv3 × 3, C(8, 3, 3, 1, 1) |

The detailed architecture of MobileFusion is summarized in Tab. 7. We omit the identity operation. It consists of (i) shallow cross-modal features fusion, (ii) $M = 2$ stacked lightweight blocks for deep features extraction, and (iii) a final reconstruction stage that generates the fused image. The infrared and visible inputs are concatenated along the channel dimension and passed to the network. The shallow feature fusion stage aims to capture modality-specific low-level cues (e.g., textures and edges) and to establish initial cross-modal interactions, producing an early fused representation. The deep feature extraction stage employs two cascaded RepMBConv ($3 \times 3$) layers to mine higher-level semantic information and perform more thorough integration. This process is further strengthened by a lightweight context-aware attention module that jointly reinforces global dependencies and local details, together with a re-parameterized feed-forward network that boosts nonlinear expressiveness and enhances representational capacity. MobileFusion strikes a favorable balance between performance and computational efficiency, effectively enabling real-world applications of infrared and visible image fusion, particularly in resource-constrained scenarios.

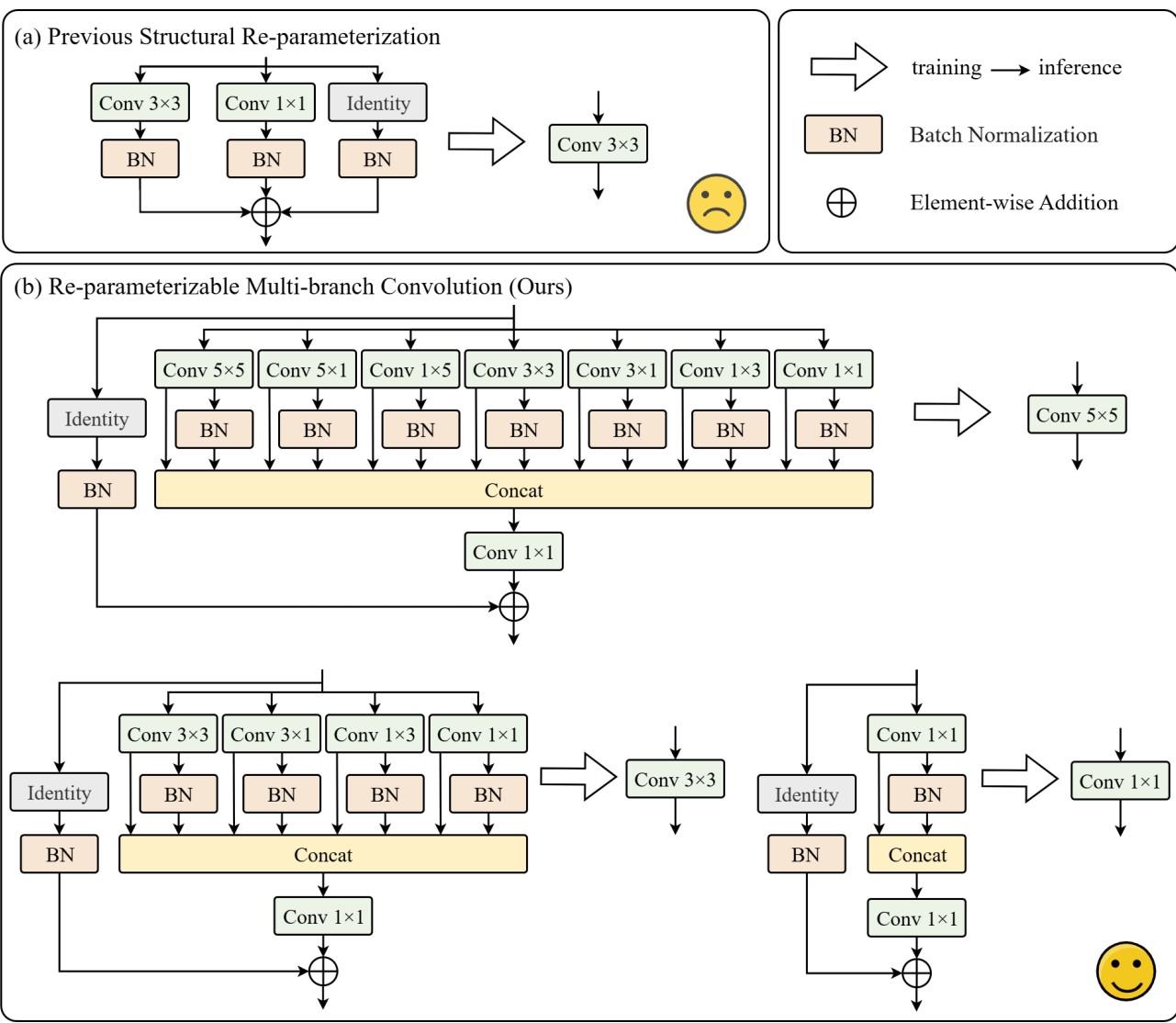

*Figure 6.* Comparison of Re-parameterizable Convolution Architectures.

## B. RepMBConv: A Fusion-Oriented Re-parameterizable Multi-Branch Convolution

Although structural re-parameterization has been widely used in efficient architecture design for high-level vision tasks (e.g., FastViT (Vasu et al., 2023a), RepVGG, MobileOne, and RepViT), which primarily focus on overall semantic recognition, it lacks a modality interaction mechanism. This limitation makes its direct application to low-level infrared and visible

image fusion ineffective. As illustrated in Fig. 5(d), features from the two modalities are neither sufficiently extracted nor effectively interacted, leading to inadequate fusion of complementary information, reduced saliency and blurred textures in the fused images. This limitation stems from the fact that conventional re-parameterization (Rep) designs focus solely on enhancing single-stream representations and do not encourage cross-modal interactions. The architectural comparison is shown in Fig. 6.

(1) **Branch heterogeneity for complementary cues:** RepMBConv incorporates heterogeneous branches with multiple kernel sizes, effectively enlarging the receptive field, enhancing multi-modal feature extraction, and facilitating cross-modal interactions, as highlighted in Tab. 6, in contrast to traditional Rep designs.

(2) **Interaction-aware inductive bias:** RepMBConv adopts a concatenation-and-projection strategy across all branches. Specifically, each branch first expands the input along the channel dimension, after which the branch outputs are concatenated and fused via a $1 \times 1$ convolution that projects the aggregated representation to the target output dimensionality. Compared with element-wise summation, this design explicitly enhances cross-modal interactions and enables more effective fusion of complementary information. We conduct a experiment on the MSRS dataset to validate the performance of replacing the concatenation and projection of RepMBConv with direct element-wise addition. This results in a decline in fusion performance, as reported in Tab. 8.

*Table 8.* Comparison between concatenation followed by projection and direct element-wise addition in RepMBConv on the MSRS dataset.

| Method | EN↑ | SCD↑ | CC↑ | SSIM↑ | MI↑ |
|---|---|---|---|---|---|
| RepMBConv (Element-wise Addition) | 6.614 | 1.658 | 0.599 | 0.926 | 2.863 |
| RepMBConv (Ours) | **6.670** | **1.766** | **0.616** | **0.956** | **3.172** |

(3) **Deployment-equivalent mapping to fusion stability:** Low-level multi-modal fusion is sensitive to intensity and contrast shifts. To improve fusion stability, RepMBConv analytically merges convolutional branches including those coupled with normalization into a single equivalent operator at inference time, ensuring stable deployment. This design helps preserve the original features while smoothing the representations and reducing sensitivity to variations in the input distribution. As reported in Tab. 9, the absence of either convolutional branches or parallel BN normalization branches leads to unstable re-parameterization deployment and degraded fusion performance.

*Table 9.* Impact of convolution and BN branches in RepMBConv on the MSRS dataset.

| Method | EN↑ | SCD↑ | CC↑ | SSIM↑ | MI↑ |
|---|---|---|---|---|---|
| RepMBConv (No Convolutional Branches) | 6.631 | 1.640 | 0.601 | 0.937 | 3.069 |
| RepMBConv (No BN Branches) | 6.537 | 1.594 | 0.563 | 0.911 | 2.895 |
| RepMBConv (Ours) | **6.670** | **1.766** | **0.616** | **0.956** | **3.172** |

Overall, extensive experiments and ablation studies validate the effectiveness of RepMBConv for infrared and visible image fusion tasks.

## C. Performance on Semantic Segmentation Task

In the semantic segmentation task, we employ Segformer (Xie et al., 2021) as the backbone and train it on the source images of FMB. Qualitative and quantitative results are presented in Fig. 7 and Tab. 10. As shown in Fig. 7, our method not only accurately segments the road in the scene but also preserves the fine contours of cars and pedestrians with greater completeness. This highlights our method's strength in preserving detailed spatial information, enabling more accurate and fine-grained segmentations. As shown in Tab. 10, our method achieves the second-best mean Intersection-over-Union (mIoU), demonstrating its excellent ability to preserve semantic information while requiring minimal computational overhead.

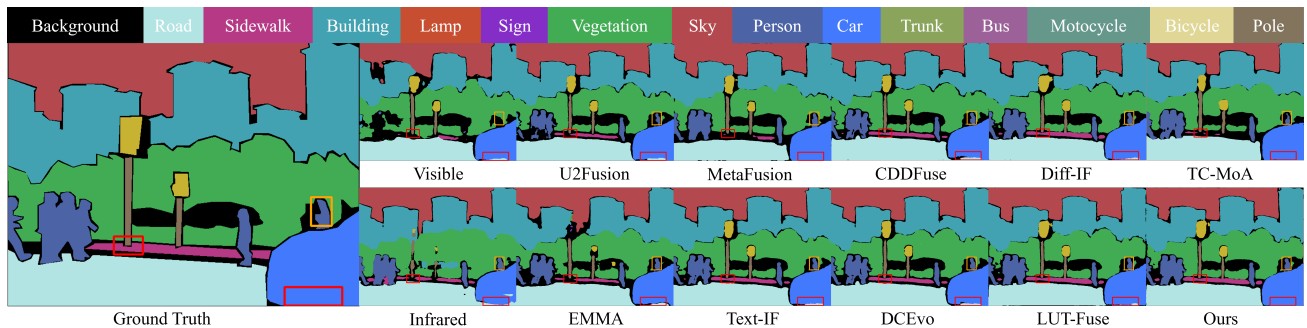

*Figure 7.* Qualitative comparison in semantic segmentation task on the FMB dataset.

*Table 10.* Quantitative comparison of semantic segmentation performance on fused images.

| Methods | Person | Road | Sign | Car | Building | mIoU |
|---|---|---|---|---|---|---|
| U2Fusion | 59.20 | 85.42 | 67.94 | 79.13 | 81.94 | 56.53 |
| MetaFusion | 59.82 | 84.83 | 66.86 | 80.25 | 81.32 | 55.42 |
| CDDFuse | 60.34 | 87.06 | 66.65 | 79.27 | 81.55 | 57.68 |
| Diff-IF | 59.75 | 86.32 | 64.62 | 79.92 | 80.97 | 56.12 |
| TC-MoA | 59.15 | 83.54 | 67.35 | 80.31 | 81.56 | 55.59 |
| EMMA | 59.84 | 85.51 | 66.52 | 78.07 | 80.02 | 56.36 |
| Text-IF | 60.81 | 87.25 | 68.17 | 79.85 | **82.13** | 57.89 |
| DCEvo | **61.46** | **87.52** | **69.34** | 79.74 | 81.79 | **58.43** |
| LUT-Fuse | 61.19 | 87.35 | 67.92 | 79.94 | 81.72 | 58.14 |
| **MobileFusion** | 61.37 | 87.13 | 68.56 | **80.55** | 81.85 | 58.22 |

## D. Generalization to Multi-modal Medical Image Fusion

To assess the generalization ability of MobileFusion beyond infrared and visible image fusion, we further evaluate it on the multi-modal medical image fusion task. We compare MobileFusion against state-of-the-art medical image fusion models or general-purpose multi-modal image fusion approaches, including U2Fusion, SwinFusion (Ma et al., 2022), CDDFuse, DDFM (Zhao et al., 2023d), TC-MoA, EMMA, and PSLPT (Wang et al., 2024b). For quantitative comparisons, we employ the metrics including MI, spatial frequency (SF), CC, SCD, and visual information fidelity (VIF). Higher values of MI, SF, CC, SCD, and VIF indicate higher quality of the fusion image. Notably, we fine-tune MobileFusion for only 16 epochs on the PET-MRI subset of the Harvard[2] medical dataset. We use 200 image pairs for fine-tuning and 20 for testing. Quantitative and qualitative results are presented in Tab. 11 and Fig. 8.

*Table 11.* Quantitative comparison on the PET-MRI dataset.

| Method | MI↑ | SF↑ | CC↑ | SCD↑ | VIF↑ |
|---|---|---|---|---|---|
| U2Fusion | 2.785 | 35.083 | 0.789 | 0.947 | 0.460 |
| SwinFusion | **3.662** | 34.294 | 0.821 | 1.642 | 0.703 |
| CDDFuse | 3.572 | **38.307** | 0.810 | **1.481** | 0.650 |
| DDFM | 3.356 | 36.257 | 0.817 | 1.413 | 0.685 |
| TC-MoA | 2.787 | 24.350 | 0.823 | 1.387 | 0.641 |
| EMMA | 3.219 | 32.887 | 0.794 | 1.156 | 0.635 |
| PSLPT | 3.612 | 35.984 | 0.803 | 0.986 | **0.713** |
| **MobileFusion** | 3.582 | 37.618 | **0.826** | 1.425 | 0.669 |

---

[2] https://www.med.harvard.edu/AANLIB/home.html

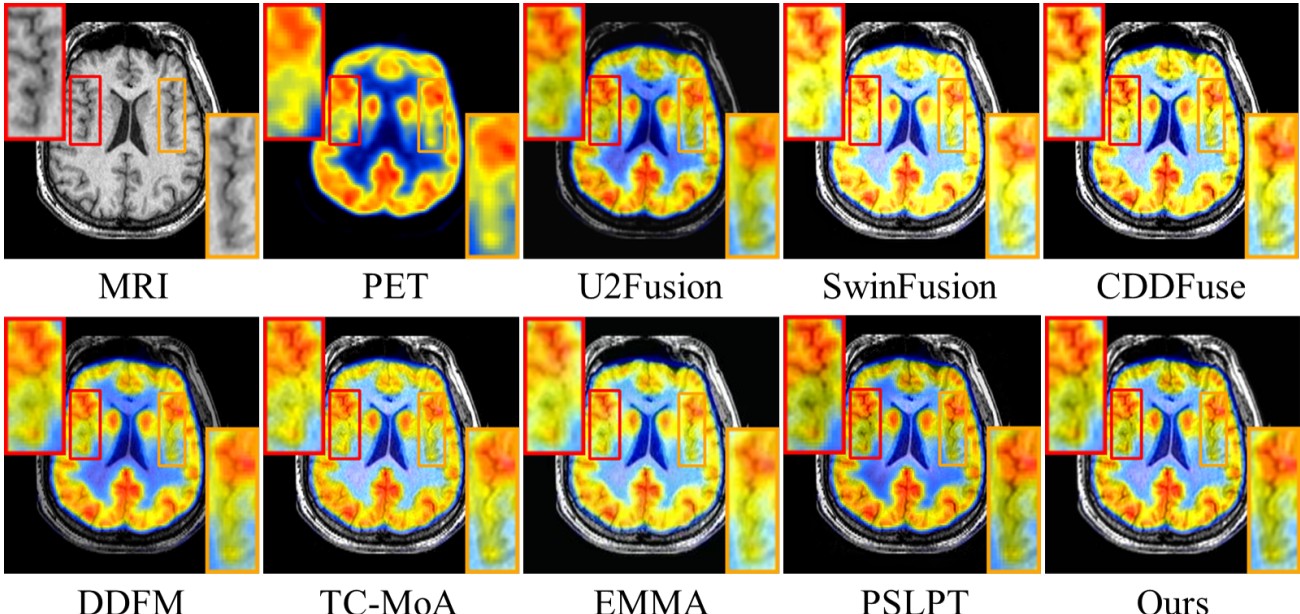

*Figure 8.* Qualitative comparison on the PET-MRI dataset.

MobileFusion achieves competitive results by effectively extracting critical pathological structures and fine-grained texture features, and seamlessly integrating them into the fused medical images, highlighting its robust cross-domain transferability. This effectiveness stems from MobileFusion's key design elements: (i) the multi-scale structural modeling enabled by RepMBConv, (ii) the lightweight context-aware enhancement module, which jointly preserves global semantics and accentuates subtle pathological structures critical for clinical diagnosis, and (iii) a re-parameterized feed-forward network that enhances nonlinearity and improves robustness. Most importantly, thanks to its optimized re-parameterizable architecture and extremely low parameter count, MobileFusion achieves this high fidelity with significantly reduced computational complexity and memory footprint. This efficiency makes it particularly well-suited for deployment on resource-constrained medical devices (e.g., portable scanners and intraoperative imaging systems), where real-time performance and efficiency are stringent requirements in clinical practice.

## E. More Ablation Experiments

To further validate the design rationale of MobileFusion, we conduct additional ablation studies on the internal components of LCAA and the kernel-structure design of RepMBConv. In addition, we analyze the expansion ratio $r$ of RepFFN. These results further support the effectiveness of our architectural designs.

First, as reported in Tab. 12, the component-wise ablation of LCAA on FMB shows that the complete LCAA consistently outperforms its individual channel-only and spatial-only variants. The dual-stream channel attention mainly contributes to modality-specific channel recalibration, helping suppress redundant responses and enhance informative infrared-visible cues. In contrast, the dual-direction spatial attention is more effective in modeling horizontal and vertical contextual dependencies, thereby emphasizing local edges, contours, and structural details. When these two components are combined, LCAA achieves the best performance across all metrics, indicating that channel-wise global statistics and directional spatial cues play complementary roles in infrared-visible image fusion.

*Table 12.* Ablation study of the proposed LCAA module on the FMB dataset.

| Method | EN↑ | SCD↑ | CC↑ | SSIM↑ | MI↑ |
|---|---|---|---|---|---|
| Dual-stream channel attention | 6.642 | 1.548 | 0.613 | 0.970 | 2.289 |
| Dual-direction spatial attention | 6.635 | 1.552 | 0.602 | 0.965 | 3.284 |
| LCAA | **6.685** | **1.601** | **0.623** | **0.986** | **3.299** |

Second, as reported in Tab. 13, the kernel-structure sensitivity analysis of RepMBConv on MSRS demonstrates the importance of heterogeneous kernel geometries. A single asymmetric branch such as $1 \times k$ provides limited receptive-field coverage and yields relatively inferior performance. The standard $k \times k$ branch improves overall feature aggregation and better captures broader thermal target distributions. Moreover, adding asymmetric $1 \times k$ or $k \times 1$ branches further improves fusion quality by enhancing the modeling of horizontal and vertical visible structures. The full design, i.e., $k \times k + 1 \times k + k \times 1$, achieves the best overall results, confirming that diverse kernel geometries provide a stronger inductive bias for cross-modal feature interaction and structural detail preservation.

*Table 13.* Kernel-structure sensitivity analysis of RepMBConv on the MSRS dataset.

| Method | EN↑ | SCD↑ | CC↑ | SSIM↑ | MI↑ |
|---|---|---|---|---|---|
| $1 \times k$ | 6.173 | 1.246 | 0.441 | 0.794 | 2.852 |
| $k \times k$ | 6.417 | 1.495 | 0.583 | 0.910 | 2.976 |
| $k \times k + 1 \times k$ | 6.521 | 1.612 | 0.598 | 0.932 | 3.041 |
| $k \times k + k \times 1$ | 6.588 | 1.617 | 0.596 | 0.945 | 3.102 |
| $k \times k + 1 \times k + k \times 1$ | **6.670** | **1.766** | **0.616** | **0.956** | **3.172** |

Third, as reported in Tab. 14, the ablation on the expansion ratio $r$ of RepFFN on the FMB dataset reveals a clear trade-off between nonlinear representation capacity and deployment efficiency. Compared with $r = 1$, setting $r = 2$ consistently improves EN, SCD, CC, SSIM, and MI. This indicates that a moderate expansion of the intermediate channels provides RepFFN with a richer hidden feature space for nonlinear transformation. Specifically, the expanded representation allows the PReLU activation between two $1 \times 1$ RepMBConv layers to perform more effective nonlinear channel mixing, which helps further integrate infrared saliency, visible texture details, and structural cues after LCAA refinement. As a result, the fused representation can better preserve complementary information from both modalities and achieve improved structural consistency. Although $r = 4$ further improves the quantitative metrics, it increases the latency from 2.10 ms to 3.92 ms, introducing approximately 87% higher latency. Since MobileFusion targets real-time deployment on resource-constrained platforms, we adopt $r = 2$ as the default setting to achieve a favorable balance between fusion quality and computational efficiency.

*Table 14.* Ablation study on the expansion ratio $r$ of RepFFN on the FMB dataset.

| Method | EN↑ | SCD↑ | CC↑ | SSIM↑ | MI↑ | Latency (ms)↓ |
|---|---|---|---|---|---|---|
| $r = 1$ | 6.674 | 1.588 | 0.612 | 0.977 | 3.289 | **1.63** |
| $r = 2$ | 6.685 | 1.601 | 0.623 | 0.986 | 3.299 | 2.10 |
| $r = 4$ | **6.735** | **1.702** | **0.684** | **0.989** | **3.426** | 3.92 |

