# OpenReview forum: "MobileFusion: Mobile-Friendly Infrared and Visible Image Fusion via Structural Re-parameterization"
_ICML.cc/2026/Conference — ICML 2026 regular_

### Official Review · Reviewer_vEnX · 2026-03-05

**Soundness:** 3
**Presentation:** 3
**Significance:** 3
**Originality:** 2
**Overall Recommendation:** 4
**Confidence:** 3

**Summary:**

This paper presents MobileFusion, an extremely lightweight and effective convolutional framework designed for the infrared and visible image fusion (IVIF) task.
MobileFusion leverages a Re-parameterizable Multi-branch Convolution (RepMBConv) module, which promotes synergy between the two modalities during training and collapses into a single-path operator to enable high-speed inference. To further enhance fusion quality, the framework incorporates a Lightweight Context-Aware Attention (LCAA) module to capture global context and a re-parameterized Feed-Forward Network (RepFFN) to strengthen non-linear feature expressiveness.
The proposed method achieves state-of-the-art  performance in both IVIF and various downstream tasks while maintaining the efficiency required for real-time inference.

**Compliance With Llm Reviewing Policy:**

Affirmed.

**Final Justification:**

The rebuttal effectively demonstrates the individual contribution of each LCAA module and the advantages of utilizing diverse kernels. However, I still find the technical novelty of RepFFN to be limited, and the concern regarding W.4 (Performance Saturation) remains unresolved. Furthermore, the authors did not address my previous point that "the empirical validation of cross-modal synergy is confined to infrared and visible image pairs." For a paper focusing on modality fusion, the evaluation across such limited modalities may be insufficient. Nonetheless, recognizing the significant practical merits of the deployment-oriented efficient processing, I maintain my original score.

**Key Questions For Authors:**

What is the specific magnitude of performance improvement contributed by the Dual-stream Channel Attention and the Dual-direction Spatial Attention individually?

How would the performance change if the kernels were restricted (e.g., using only $k \times k$ and $1 \times k$ while excluding $k \times 1$)?

**Limitations:**

The paper lacks a discussion on "failure cases." Identifying specific conditions where MobileFusion might struggle compared to existing models would be highly informative.

**Strengths And Weaknesses:**

### **Strengths**

* **Clear task setting and high practical utility**: The model achieves state-of-the-art (SOTA) accuracy across multiple benchmarks and downstream tasks despite an extremely small footprint of approximately 4K parameters. It demonstrates high real-world potential by enabling real-time inference on edge platforms like the NVIDIA Jetson Orin NX.
* **Technically sound architecture**: The use of re-parameterizable multiple branches with heterogeneous kernel geometries, such as $k \times k$, $k \times 1$, and $1 \times k$, is a technically grounded approach. This design maintains representational capacity during training while collapsing into an efficient single-path operator for deployment.
* **Comprehensive experimental validation**: The authors conduct extensive experiments across standard IVIF benchmarks and high-level downstream tasks, including object detection and semantic segmentation.

---

### **Weaknesses**

* **Lack of theoretical grounding and limited domain validation**: While the authors claim this design provide a stronger inductive bias for modality interactions, there is no explicit theoretical guarantee provided for why kernel size diversity specifically enhances cross-modal fusion. Also, It is unclear whether the performance gains of RepMBConv are primarily driven by the diversity of kernel sizes or the strategy of concatenating multi-branch outputs. Consequently, the study lacks sufficient empirical validation to determine if diverse kernel geometries are truly crucial for the fusion process. Furthermore, the empirical validation of cross-modal synergy is confined to infrared and visible image pairs.
* **Insufficient ablation of LCAA components**: The ablation studies evaluate the Lightweight Context-Aware Attention (LCAA) only as a complete module. The individual effectiveness and necessity of combining Dual-Stream Channel Attention with Dual-Direction Spatial Attention are not isolated or demonstrated through evaluation.
* **Limited novelty and impact of RepFFN**: The Re-parameterized Feed-Forward Network (RepFFN) follows a standard two-stage $1 \times 1$ convolutional architecture. Quantitative results in the ablation table show that its contribution to overall performance is marginal, limiting its technical impact.
* **Performance saturation and marginal impact compared to baselines**: Quantitative results across many tables appear saturated, with only limited accuracy gains over the existing real-time baseline, LUT-Fuse. Given that LUT-Fuse already established a significant breakthrough in inference speed, the incremental impact of this work is relatively constrained.

---

> ### Author Rebuttal · Authors · 2026-03-31
>
> We sincerely appreciate the reviewer’s positive feedback.
>
> Overall response: Our goal is not to claim a formal theoretical guarantee, but to provide a lightweight, deployment-oriented IVIF-specific design with a strong quality-efficiency trade-off under strict resource constraints.
>
> Q1: The channel attention mechanism helps suppress redundancy and reduce noise, whereas the subsequent spatial attention mechanism more effectively emphasizes local edges and structural details. We have added component-wise ablation studies for LCAA on the FMB dataset, and the results clearly demonstrate that the two components play complementary roles and achieve the best performance when used together.
> | Method | EN | SCD | CC | SSIM | MI |
> | :----: | :--: | :--: | :--: | :--: | :--: |
> | dual-stream channel attention | 6.642 | 1.548 | 0.613 | 0.970 | 2.289 |
> | dual-direction spatial attention | 6.635 | 1.552 | 0.602 | 0.965 | 3.284 |
> | LCAA | 6.685 | 1.601 | 0.623 | 0.986 | 3.299 |
>
> Q2: We have added a kernel-structure sensitivity analysis for RepMBConv on the MSRS dataset.
> | Method | EN | SCD | CC | SSIM | MI |
> | :----: | :--: | :--: | :--: | :--: | :--: |
> | 1 $\times$ k | 6.173 | 1.246 | 0.441 | 0.794 | 2.852 |
> | k $\times$ k | 6.417 | 1.495 | 0.583 | 0.910 | 2.976 |
> | k $\times$ k + 1 $\times$ k | 6.521 | 1.612 | 0.598 | 0.932 | 3.041 |
> | k $\times$ k + k $\times$ 1 | 6.588 | 1.617 | 0.596 | 0.945 | 3.102 |
> | k $\times$ k + 1 $\times$ k + k $\times$ 1 | 6.670 | 1.766 | 0.616 | 0.956 | 3.172 |
>
> The k $\times$ k branch helps model broader thermal distributions, while the asymmetric 1 $\times$ k and k $\times$ 1 branches are important for capturing horizontal and vertical visible structures. The full heterogeneous design consistently performs best, supporting the necessity of diverse kernel geometries.
>
> Q3: Our point is not to pursue a large absolute margin over every real-time baseline, but to achieve strong fusion quality under an ultra-small model budget. We further scale the model to a MobileFusion-Large variant by specifically increasing the base channel width from 8 to 12 and the number of stacked lightweight blocks from 2 to 4, while keeping all other settings unchanged. This scaled-up version achieves comprehensive state-of-the-art performance, demonstrating the effectiveness and scalability of the proposed architecture. The standard and large variants also provide flexible trade-offs among inference speed, model size, and fusion quality. We will include the Large version in the revised manuscript.
> | Method | EN | SCD | CC | SSIM | MI | Params(M) | Latency(ms) |
> | :----: | :--: | :--: | :--: | :--: | :--: | :-------: | :---------: |
> | MobileFusion | 6.685 | 1.601 | 0.623 | 0.986 | 3.299 | 0.004 | 2.10 |
> | MobileFusion-Large | 6.748 | 1.712 | 0.703 | 0.991 | 3.528 | 0.030 | 5.07 |
>
> Q4: The role of RepFFN is to provide low-cost nonlinear enhancement after fusion-oriented interaction, while its parameter budget and enhancement strength are controlled by the expansion ratio. As reported in Reviewer WwsM-Q4, when the expansion ratio is increased from r = 2 to r = 4, the EN, CC, and MI metrics improve noticeably. However, this also leads to an approximately 87% increase in inference latency compared with the original setting. Since our goal is to achieve high-quality fusion under an extremely strict efficiency budget, with particular emphasis on fast inference, we adopt r = 2 as the expansion ratio for RepFFN in our final design.
>
> Q5: MobileFusion may struggle in severely noisy or degraded scenarios, such as snow, rain, and heavy fog, since these conditions can simultaneously damage thermal saliency and fine-grained texture cues. Compared with IVIF methods specifically designed for image enhancement, our method is less effective at removing such adverse factors. Nevertheless, in noisy environments, our method shows more stable performance than the lightweight baseline LUT-Fuse. We add quantitative robustness experiments for LUT-Fuse and MobileFusion under Gaussian sensor noise with a standard deviation of σ = 0.05. The noise robustness evaluation results on the FMB dataset are as follows.
> | Method | EN | SCD | CC | SSIM | MI |
> | :----: | :--: | :--: | :--: | :--: | :--: |
> | LUT-Fuse | 5.842 ↓11.7% | 1.186 ↓21.8% |0.498 ↓17.8% | 0.903 ↓7.5% | 2.512 ↓25.7% |
> | MobileFusion | 6.358 ↓4.9% | 1.492 ↓6.8% | 0.592 ↓5.0% | 0.958 ↓2.8% | 3.097 ↓6.1% |
>
> The limitation do not contradict the core objective of MobileFusion. Our goal is to achieve a better balance among fusion quality, model compactness, and real-time edge efficiency. Rather, these failure cases reflect the practical performance boundary of a lightweight fusion architecture that does not explicitly model strong restoration capability. In the revised manuscript, we will include representative failure cases, compare them with existing methods, and provide a clearer discussion of the corresponding limitations and future directions.

---

> > ### Author Rebuttal · Reviewer_vEnX · 2026-04-02
> >
> > I would like to thank the authors for their rebuttal. The response effectively demonstrates the individual contribution of each LCAA module and the advantages of utilizing diverse kernels. However, I still find the technical novelty of RepFFN to be limited, and the concern regarding W.4 (Performance Saturation) remains unresolved.
> >
> > Furthermore, the authors did not address my previous point that "the empirical validation of cross-modal synergy is confined to infrared and visible image pairs." For a paper focusing on modality fusion, the evaluation across such limited modalities may be insufficient. Nonetheless, recognizing the significant practical merits of the deployment-oriented efficient processing, I maintain my original score.

---

> > > ### Author Response · Authors · 2026-04-05
> > >
> > > We sincerely appreciate the reviewer's continued positive feedback.
> > >
> > > We sincerely apologize for not explicitly addressing this point "the empirical validation of cross-modal synergy is confined to infrared and visible image pairs." in our initial rebuttal. Thank you very much for your follow-up feedback.
> > >
> > > Q1: Although our work primarily focuses on infrared-visible image fusion, the modular design of MobileFusion is inherently generalizable and readily adaptable to other multi-modal image fusion tasks. We provided extensive experiments in the Appendix (Section D: Generalization to Multi-modal Medical Image Fusion) to demonstrate MobileFusion’s strong transferability beyond infrared-visible image pairs. Remarkably, with only a few fine-tuning steps and without training from scratch on medical data, MobileFusion achieves performance competitive with state-of-the-art methods specifically designed and trained for medical image fusion. Furthermore, we have added results on digital photographic image fusion on the Lytro dataset, as shown below. MobileFusion attains higher CC and MI scores than GIFNet (CVPR 2025), a general-purpose fusion model specifically trained on the Lytro dataset, while remaining strongly competitive on the other metrics as well.
> > > | Method | EN | SCD | CC | SSIM | MI |
> > > | :-----------: | :------------: | :------------: | :------------: | :------------: | :------------: |
> > > | GIFNet | 7.575 | 1.370 | 0.973| 0.899 | 6.426|
> > > | LUT-Fuse | 6.226 | 1.146 | 0.942 | 0.862 | 5.947 |
> > > | MobileFusion | 7.374 | 1.315 | 0.981 | 0.880 | 6.587 |
> > >
> > > Overall, MobileFusion is not limited to infrared-visible image fusion. Instead, it captures general cross-modal interaction patterns and serves as a general, efficient, and adaptable framework for multi-modal image fusion across diverse application scenarios. Notably, with only about 4K parameters, it is well suited for flexible deployment on edge devices while maintaining high fusion quality.
> > >
> > > Q2-(Performance Saturation): We agree that the absolute gains over LUT-Fuse in terms of SSIM and CC are modest. However, the value of MobileFusion lies in its ability to achieve fusion performance that is comprehensively competitive with heavy Transformer- or diffusion-based models, but in a highly efficient and ultra-lightweight setting designed for strict real-time constraints. Compared to LUT-Fuse, MobileFusion is significantly smaller, faster, and more robust, particularly under noise, resolution scaling, and moderate misalignment pressures, as detailed in Reviewer S7wR-Q3. This demonstrates its stable and reliable applicability in real-world scenarios without catastrophic failure.
> > >
> > > Q3-(Technical novelty of RepFFN): RepFFN is a lightweight feed-forward module with a reparameterized structure. While it does not introduce a fundamentally novel operator when considered independently, it is designed to cooperate synergistically with the preceding feature extraction branches and boost the model’s nonlinear representation capacity in an efficient manner. By placing it at the end of each basic block, we enhance the model’s nonlinear fitting ability with negligible extra computation. Increasing its expansion ratio from r=2 to r=4 yields noticeable improvements in EN, CC, and MI metrics, but at the cost of approximately 87% higher inference latency (Reviewer WwsM-Q4).

---

### Official Review · Reviewer_WwsM · 2026-03-08

**Soundness:** 2
**Presentation:** 2
**Significance:** 2
**Originality:** 1
**Overall Recommendation:** 4
**Confidence:** 4

**Summary:**

This manuscript proposes MobileFusion, an ultra-lightweight convolutional framework for infrared and visible image fusion designed specifically for resource-constrained mobile devices. To break through computational bottlenecks, the model introduces structural re-parameterization technology. During the training phase, RepMBConv is utilized to promote cross-modal feature interaction. During the inference phase, this module is equivalently folded into a single-path operator, significantly reducing latency and streamlining the structure. Furthermore, the model combines LCAA to capture global and local details and introduces RepFFN to enhance nonlinear expression. The overall goal of the network is to strike a balance between fusion quality and inference speed.

**Compliance With Llm Reviewing Policy:**

Affirmed.

**Final Justification:**

The authors have addressed all my concerns in the rebuttal, and I would like to increase my rating.

**Key Questions For Authors:**

(1)	The data in Table 1 shows that, compared to other baseline methods, the improvement of MobileFusion on core fusion metrics such as SSIM and CC is extremely marginal, and it does not even possess an absolute advantage in certain metrics. The authors should provide statistical significance tests and further demonstrate how this marginal performance gain truly translates into a usability advantage in actual mobile downstream tasks.
(2)	Structural re-parameterization is already highly mature in high-level vision tasks (e.g., RepVGG, MobileOne). Besides adding different convolutional kernel sizes in RepMBConv to fuse features, what unique and fundamental theoretical innovation does this architecture design offer tailored to the specific physical characteristics of infrared and visible modalities (such as thermal radiation distribution and texture frequency differences)?
(3)	Considering that real-world mobile deployment environments are usually accompanied by severe sensor noise or image alignment errors, how does the extremely lightweight RepMBConv ensure the model's noise resistance and robustness under harsh conditions after discarding complex feature extraction mechanisms? Have relevant noise-injection tests been conducted?
(4)	In the RepFFN, the channel expansion ratio is hardcoded to r=2. How was this hyperparameter derived through rigorous search or ablation?

**Limitations:**

Investigating the balance between inference and performance in the IVIF field is highly valuable, and the manuscript tries to demonstrate the advantages of the proposed method. However, the manuscript still has shortcomings in terms of theoretical proof, parameter settings, and model performance. Currently, I consider that this manuscript is not yet sufficient for publication at ICML, and the authors still need to comprehensively improve the manuscript.

**Strengths And Weaknesses:**

The research significance of this manuscript is clear, and its innovative RepMBConv aims to compensate for the insufficient feature expression of lightweight architectures; aside from some non-standard writing, the overall logic of the paper is clear. However, based on the experimental results, the improvement of its core fusion metrics (such as SSIM and MI) compared to some baseline models is extremely marginal, lacking proof of statistical significance. Secondly, the combination of re-parameterization and attention mechanisms is already common in high-level vision, making its direct transfer to multi-modal fusion slightly thin in terms of underlying theoretical innovation. Finally, the experiments lack a robustness ablation analysis against harsh real-world mobile conditions (such as severe noise and misalignment), which weakens the comprehensiveness of the conclusions.

---

> ### Author Rebuttal · Authors · 2026-03-31
>
> We sincerely appreciate the reviewer’s positive feedback.
>
> Overall response: Our contribution is not a fundamental new theory for infrared-visible imaging, but a lightweight, deployment-oriented IVIF-specific design that seeks a strong quality-efficiency trade-off under strict resource constraints.
>
> Q1: We agree that the gains in several metrics, such as SSIM and CC, are relatively modest, and statistical significance analysis is necessary for a more rigorous comparison. In the revised manuscript, we will therefore include repeated-run experiments together with significance tests to better quantify the reliability of the observed improvements. More importantly, our main claim is not that MobileFusion achieves the absolute best score on every individual metric, but that it provides a more favorable trade-off between fusion quality, model compactness, and real-time edge latency. In other words, the practical value of MobileFusion lies in maintaining competitive fusion performance under extremely limited parameter and deployment budgets, which is more relevant for mobile scenarios than pursuing marginal gains in isolated full-reference metrics alone. In addition, to verify that the proposed design is not limited to an ultra-lightweight setting, we also construct a larger variant by increasing the base channel width from 8 to 12 and the number of stacked lightweight blocks from 2 to 4, while keeping the overall architecture unchanged. This larger model achieves the best results across all metrics on the FMB dataset, which further confirms the effectiveness and scalability of our design. These results also show that our architecture can flexibly support different resource–performance trade-offs, allowing users to choose an appropriate model size according to practical deployment requirements. We will include these additional results in the revised manuscript. The FMB evaluation results on an NVIDIA GeForce RTX 4090 are as follows:
> | Method | EN | SCD | CC | SSIM | MI | Params(M) | Latency(ms) |
> | :----: | :--: | :--: | :--: | :--: | :--: | :-------: | :---------: |
> | MobileFusion | 6.685 | 1.601 | 0.623 | 0.986 | 3.299 | 0.004 | 2.10 |
> | MobileFusion-Large | 6.748 | 1.712 | 0.703 | 0.991 | 3.528 | 0.030 | 5.07 |
>
> Q2: In contrast to conventional reparameterizable modules that focus on intra-modal representation for classification, our RepMBConv is explicitly designed to address the physical heterogeneity between infrared and visible modalities. Infrared images convey low-frequency thermal saliency, while visible images provide high-frequency textures and structural details. Standard Rep-style blocks, when applied directly to fusion tasks, often process both modalities homogeneously and fail to preserve or integrate their distinct semantics. RepMBConv overcomes this by introducing a fusion-aware heterogeneous multi-branch inductive bias: parallel branches with diverse receptive fields and directional sensitivities extract complementary cross-modal cues, and a lightweight projection layer adaptively aggregates them into a unified representation. This design enables explicit modeling of thermal saliency, structural consistency, and textural complementarity during training, while remaining fully reparameterizable into a compact form for efficient deployment. We provide a detailed technical explanation of this fusion-oriented architecture in Reviewer z81f-Q1 and Reviewer S7wR-Q2.
>
> Q3: We provide quantitative robustness experiments for LUT-Fuse and MobileFusion under Gaussian sensor noise with a standard deviation of σ = 0.05. LUT-Fuse employs a lookup-table-based architecture, which suffers from unstable performance under noisy conditions due to its fixed, non-adaptive mapping that lacks robustness to input perturbations. In contrast, our MobileFusion enhances feature representation through a context-aware attention mechanism and the nonlinear capacity of RepFNN, thereby significantly improving stability and robustness in noise environments. The noise robustness evaluation results on the FMB dataset are as follows.
> | Method | EN | SCD | CC | SSIM | MI |
> | :----: | :--: | :--: | :--: | :--: | :--: |
> | LUT-Fuse | 5.842 ↓11.7% | 1.186 ↓21.8% |0.498 ↓17.8% | 0.903 ↓7.5% | 2.512 ↓25.7% |
> | MobileFusion | 6.358 ↓4.9% | 1.492 ↓6.8% | 0.592 ↓5.0% | 0.958 ↓2.8% | 3.097 ↓6.1% |
>
> Q4: The choice of r=2 was made to balance nonlinear enhancement and the tight efficiency budget on mobile devices. Ablations with r=1, r=2, r=4 show that r=2 improves nonlinear capacity without introducing excessive overhead. We will add this ablation more explicitly in the revised manuscript.
> | Method | EN | SCD | CC | SSIM | MI | Params(M) | Latency(ms) |
> | :----: | :--: | :--: | :--: | :--: | :--: | :-------: | :---------: |
> | r=1 | 6.674 | 1.588 | 0.612 | 0.977 | 3.289 | 0.002 | 1.63 |
> | r=2 | 6.685 | 1.601 | 0.623 | 0.986 | 3.299 | 0.004 | 2.10 |
> | r=4 | 6.735 | 1.702 | 0.684 | 0.989 | 3.426 | 0.010 | 3.92 |

---

> > ### Author Rebuttal · Reviewer_WwsM · 2026-04-04
> >
> > The authors have addressed all my concerns in the rebuttal, and I would like to increase my rating.

---

> > > ### Author Response · Authors · 2026-04-05
> > >
> > > We are deeply grateful for your insightful comments and positive feedback. Thank you for your support and endorsement of our work.

---

### Official Review · Reviewer_S7wR · 2026-03-12

**Soundness:** 2
**Presentation:** 2
**Significance:** 2
**Originality:** 2
**Overall Recommendation:** 3
**Confidence:** 5

**Summary:**

This paper presents a lightweight CNN framework for infrared–visible image fusion, targeting real-time deployment under tight compute and memory constraints. The key design is structural re-parameterization. During training, the model uses multi-branch convolutional structures to improve optimization and cross-modal feature interaction. At inference time, these branches are analytically merged into single-path convolutions for faster execution. The architecture is built around a re-parameterizable multi-branch convolution module, augmented with a lightweight context-aware attention block, and a re-parameterized feed-forward module. Experiments report good fusion-quality metrics and fast inference on both a desktop GPU and a Jetson Orin NX edge platform.

**Compliance With Llm Reviewing Policy:**

Affirmed.

**Final Justification:**

The rebuttal and follow-up substantially improve clarity and add useful evidence. These additions strengthen the empirical story. However, my key concern remains the novelty positioning. Prior IVIF work such as SIFusion applies structural re-parameterization, so any "first" claim should be narrowed or qualified, and the novelty claim remains partially verified without a comparable evaluation. In addition, the new stress tests are compared against LUT-Fuse only. Including learning-based baselines under the same setting would make the robustness more convincing. I maintain my rating.

**Key Questions For Authors:**

1. How does MobileFusion differ technically from prior methods such as FECFusion, LDRepFM, and SIFusion, and empirical comparisons should be provided or a concrete reason they are not comparable under the same settings?

2. Beyond adopting structural re-parameterization, what is the core fusion-specific insight that cannot be obtained by directly applying existing rep-style backbones (e.g., RepVGG/MobileOne) with standard IVIF training objectives?

3. How does MobileFusion perform on a smartphone platform? If that is out of scope, what evidence supports the claim that Jetson Orin NX is a strong proxy for mobile IVIF deployment?

4. What are the exact inference settings for all reported latency numbers, including precision, runtime stack, batch size, warm-up iterations, and input resolution?

5. Are there failure cases where the extremely small model capacity struggles, such as heavy misalignment between IR/VI, extreme noise, or very high-resolution inputs?

**Limitations:**

See weaknesses.

Minor issues:
- “MobileFusion” is written as “Mobile-Fusion" in Table 4 and should be kept as the same.

**Strengths And Weaknesses:**

Strengths:

+ The paper targets a practical deployment pain point in IVIF, and the efficiency goal is clear and well motivated

+ The overall design is technically coherent, and the re-parameterization derivation is clearly presented and easy to follow


Weaknesses

- The claim that it is the first time structural re-parameterization is used in multimodal image fusion appears overstated. There are prior IVIF works that already incorporate re-parameterization ideas, e.g., FECFusion, LDRepFM, or SIFusion. Also, the paper does not compare against these baselines empirically, making it unclear how MobileFusion performs relative to existing re-parameterization-based fusion approaches.

- The technical contribution is the IVIF-oriented design of established efficiency patterns, e.g., multi-branch re-parameterizable convolutions, lightweight attention, and 1×1 expansion/FFN-like blocks. The paper does a clean systemization, but it is not clear what new learning principle or new fusion-specific mechanism is introduced beyond adopting re-parameterization for this task.

- The mobile evaluation is primarily on Jetson Orin NX. It would strengthen the claim to include one smartphone deployment, or provide a clearer argument why Jetson Orin NX is representative of the intended deployment.

- Efficiency reporting primarily focuses on latency, while power/energy and key implementation details (e.g., precision, TensorRT/ONNX export, and warm-up settings) are not fully characterized, which is also important for low-power deployment scenarios.

[1] FECFusion: Infrared and visible image fusion network based on fast edge convolution, 2023

[2] LDRepFM: A Real-time End-to-End Visible and Infrared Image Fusion Model Based on Layer Decomposition and Re-parameterization, 2023

[3] SIFusion: Lightweight infrared and visible image fusion based on semantic injection, 2024

---

> ### Author Rebuttal · Authors · 2026-03-31
>
> We sincerely appreciate the reviewer’s positive feedback.
>
> Overall response: Our contribution is not that MobileFusion is the earliest use of any re-parameterization idea in IVIF. Rather, the contribution lies in a fusion-oriented and deployment-aware lightweight design for resource-constrained IVIF deployment under strict efficiency constraints. In particular, MobileFusion centers on RepMBConv, a fusion-oriented re-parameterizable block with heterogeneous branches and concatenation-then-projection aggregation.
>
> Q1: We emphasize that MobileFusion is designed to explore a streamlined topology with hardware-friendly operators to achieve extremely fast yet high-quality fusion on mobile devices, featuring a network architecture that is significantly more suitable for edge deployment than FECFusion, LDRepFM, SIFusion, and other compared methods. More importantly, the key distinction from existing re-parameterization-based IVIF approaches lies not in merely introducing structural re-parameterization into IVIF, but in re-designing it specifically for multi-modal fusion tasks. Our goal is not only to reduce computational cost, but to deliver excellent fusion quality under strict resource constraints. We have further supplemented comparative experiments with FECFusion and LDRepFM on the FMB dataset on an NVIDIA GeForce RTX 4090, and the results clearly demonstrate that MobileFusion achieves significant advantages in both computational efficiency and fusion performance.
> | Method | EN | SCD | CC | SSIM | MI | Params(M) | Latency(ms) |
> | :----: | :--: | :--: | :--: | :--: | :--: | :-------: | :---------: |
> | FECFusion | 6.672 | 1.527 | 0.610 | 0.825 | 2.623 | 0.140 | 52.60 |
> | LDRepFM | 7.016 | 1.568 | 0.580 | 0.869 | 2.468 | 0.195 | 11.57 |
> | MobileFusion | 6.685 | 1.601 | 0.623 | 0.986 | 3.299 | 0.004 | 2.10 |
>
> Q2: Conventional reparameterizable architectures such as RepVGG are designed for single-modality classification tasks with the aim of enhancing intra-modal representational capacity, and when directly applied to IVIF under standard fusion objectives, they lack the ability to differentiate, preserve, or synergistically integrate modality-specific semantics, often leading to blurred or suppressed critical cues such as thermal saliency from infrared images and textural details from visible images; in contrast, our RepMBConv introduces a fusion-aware reparameterization paradigm specifically tailored for multi-modal fusion, as detailed in Reviewer z81f-Q1.
>
> Q3: Smartphone deployment is beyond the current scope of this work. Our target scenario is resource-constrained edge deployment, and Jetson Orin NX is used as a representative embedded edge platform to validate real-time feasibility under constrained computational budgets.
>
> Q4: Latency measurements are obtained under FP32 precision, with batch size 1, 30 warm-up iterations, and platform-specific input resolutions: 800×600 on PC and 640×480 on NVIDIA Jetson Orin NX. The runtime stack is PyTorch 2.1 with CUDA 11.8 for the PC (using native inference without additional optimization), and TensorRT for the Jetson Orin NX (where the model is first exported from PyTorch to ONNX and then converted into a TensorRT engine for efficient edge execution).
>
> Q5: Yes. Severe misalignment between infrared and visible images, extreme noise, and ultra-high-resolution inputs are all challenging scenarios, especially when the model capacity is intentionally kept small and geometric alignment is not explicitly modeled. We will include representative failure cases in the revised manuscript. We additionally provide quantitative robustness experiments for LUT-Fuse and MobileFusion under Gaussian sensor noise with a standard deviation of σ = 0.05. LUT-Fuse employs a lookup-table-based architecture, which suffers from unstable performance under noisy conditions due to its fixed, non-adaptive mapping that lacks robustness to input perturbations. In contrast, our MobileFusion enhances feature representation through a context-aware attention mechanism and the nonlinear capacity of RepFNN, thereby significantly improving stability and robustness in noise environments. The noise robustness evaluation results on the FMB dataset are as follows.
> | Method | EN | SCD | CC | SSIM | MI |
> | :----: | :--: | :--: | :--: | :--: | :--: |
> | LUT-Fuse | 5.842 ↓11.7% | 1.186 ↓21.8% |0.498 ↓17.8% | 0.903 ↓7.5% | 2.512 ↓25.7% |
> | MobileFusion | 6.358 ↓4.9% | 1.492 ↓6.8% | 0.592 ↓5.0% | 0.958 ↓2.8% | 3.097 ↓6.1% |

---

> > ### Author Rebuttal · Reviewer_S7wR · 2026-04-01
> >
> > Thanks for the rebuttal. The added measurements and comparisons are helpful. I still have a few follow-up questions to finalize my recommendation:
> >
> > - SIFusion does not appear in the added comparison. Since SIFusion also applies structural re-parameterization for IVIF, can the authors report how it performs under the same datasets and protocol (or clearly explain why it is not comparable)?
> >
> > - Can the authors add a controlled ablation where RepMBConv is replaced with a standard RepVGG/MobileOne-style re-parameterizable block, under matched parameter/FLOPs budget and the same IVIF training objective, and report the resulting performance and latency gap?
> >
> > - Beyond Gaussian noise, can the authors include additional stress tests for IR/VI misalignment and resolution scaling, which are common practical failure modes?

---

> > > ### Author Response · Authors · 2026-04-05
> > >
> > > We sincerely appreciate the reviewer's continued positive feedback.
> > >
> > > Q1: SIFusion indeed employs structural re-parameterization for IVIF. However, its official implementation is not publicly available, and the details of its network architecture and training procedure are insufficiently elaborated. To ensure evaluation fairness, we only include methods that are reproducible under the same setting. We will cite and discuss SIFusion as a relevant work in the revised manuscript. We further added inference latency comparisons on the NVIDIA Jetson Orin NX, evaluating our method against FECFusion and LDRepFM. Since our network employs a streamlined topology with hardware-friendly operators, it offers significant inference advantages on edge devices.
> > > | Method | Latency(ms) |Real-time|
> > > | :----: | :---------: |:---------: |
> > > | FECFusion | 295.30 | $\times$ |
> > > | LDRepFM | 54.79 | $\times$ |
> > > | MobileFusion | 12.88 | $\checkmark$ |
> > >
> > > Q2: We further provide a controlled ablation on an NVIDIA GeForce RTX 4090 under very closely matched parameters (0.004M) and FLOPs (1.77G) budgets and the same IVIF training objective, where RepMBConv is replaced with standard RepVGG-style and MobileOne-style reparameterizable blocks. The experimental results on the FMB dataset are as follows. The experimental results demonstrate that RepMBConv achieves significantly superior fusion performance compared to standard reparameterizable blocks. The overall framework adopts our streamlined topology, and there is no significant gap in inference speed among the three reparameterization methods. RepMBConv is specifically tailored for fusion tasks, achieving a favorable trade-off between fusion quality and computational efficiency compared to high-level reparameterization blocks.
> > > | Method | EN | SCD | CC | SSIM | MI | Latency(ms) |
> > > | :----: | :--: | :--: | :--: | :--: | :--: | :-------: |
> > > | MobileOne-style | 6.253 | 1.398 | 0.476 | 0.760 | 2.514 | 1.83 |
> > > | RepVGG-style | 6.460 | 1.419 | 0.541 | 0.873 | 2.827 | 2.15 |
> > > | RepMBConv | 6.685 | 1.601 | 0.623 | 0.986 | 3.299 | 2.10 |
> > >
> > > Q3: We add additional stress tests on the FMB dataset (800 $\times$ 600) to evaluate the robustness of our method under other two practical failure modes. For resolution scaling, we jointly downsampled ($\times$0.5) and upsampled ($\times$2) both IR and VI images using area/cubic interpolation. For spatial misalignment, we synthetically generated misaligned VI images by applying random rigid transformations (rotation: ±10°, translation: ±10 pixels) to the original aligned pairs, simulating real-world registration errors. The experimental results highlight the superior stability of MobileFusion compared to LUT-Fuse.
> > >
> > > The resolution scaling stress test (downsampled ($\times$0.5)) is conducted as follows.
> > > | Method | EN | SCD | CC | SSIM | MI |
> > > | :-----------: | :------------: | :------------: | :------------: | :------------: | :------------: |
> > > | LUT-Fuse | 6.280↓5.11% | 1.449↓4.48% | 0.592↓2.31% | 0.940↓3.69% | 3.190↓5.68% |
> > > | MobileFusion | 6.655↓0.45% | 1.591↓0.62% | 0.618↓0.80% | 0.985↓0.10% | 3.262↓1.12% |
> > >
> > > The resolution scaling stress test (upsampled ($\times$2)) is conducted as follows.
> > > | Method | EN | SCD | CC | SSIM | MI |
> > > | :-----------: | :------------: | :------------: | :------------: | :------------: | :------------: |
> > > | LUT-Fuse | 6.600↓0.27%| 1.515↓0.13% | 0.604↓0.33% | 0.972↓0.41% | 3.366↓0.47% |
> > > | MobileFusion | 6.680↓0.07%| 1.600↓0.06% | 0.622↓0.16% | 0.985↓0.09% | 3.295↓0.12% |
> > >
> > > The VI misalignment stress test is conducted as follows.
> > > | Method | EN | SCD | CC | SSIM | MI |
> > > | :-----------: | :------------: | :------------: | :------------: | :------------: | :------------: |
> > > | LUT-Fuse | 6.464↓2.33% | 1.006↓33.69% | 0.470↓22.44% | 0.753↓22.85% | 2.128↓37.08% |
> > > | MobileFusion | 6.678↓0.11% | 1.424↓11.06% | 0.581↓6.74% | 0.893↓9.43% | 2.952↓10.52% |
> > >
> > > The IR misalignment stress test is conducted as follows.
> > > | Method | EN | SCD | CC | SSIM | MI |
> > > | :-----------: | :------------: | :------------: | :------------: | :------------: | :------------: |
> > > | LUT-Fuse | 6.543↓1.13% | 1.328↓12.46% | 0.544↓10.23% | 0.903↓7.48% | 2.747↓18.78% |
> > > | MobileFusion | 6.682↓0.05% | 1.516↓5.31% | 0.592↓4.98% | 0.955↓3.14% | 3.034↓8.03% |
> > >
> > > Overall, LDRepFM, SIFusion, and FECFusion directly adopt re-parameterization techniques originally developed in high-level vision, without introducing task-specific adaptations for image fusion. Regarding our claim of being " the first to apply re-parameterization to multi-modal fusion" we will revise this statement in the revised manuscrip to make it more precise, namely, that "our work is the first to leverage novel re-parameterization techniques tailored for achieving both real-time and high-quality multi-modal fusion on edge devices".
> > >
> > > In the stress tests, MobileFusion shows better robustness and greater practical value for real-world deployment in resource-constrained scenarios than LUT-Fuse.

---

### Official Review · Reviewer_z81f · 2026-03-12

**Soundness:** 3
**Presentation:** 3
**Significance:** 3
**Originality:** 3
**Overall Recommendation:** 5
**Confidence:** 5

**Summary:**

This paper introduces MobileFusion, a lightweight convolutional framework for infrared and visible image fusion designed for mobile deployment. It utilizes a Re-parameterizable Multi-branch Convolution, paired with context-aware attention and feed-forward modules, to deliver real-time inference and high-quality fusion.

**Compliance With Llm Reviewing Policy:**

Affirmed.

**Final Justification:**

The authors clearly addressed my all concerns, which strengthens the paper’s technical clarity and overall contribution.

Clarification on Cross-Modal Interaction: The authors clearly explain how RepMBConv differs from conventional Rep designs: its channel-expanded heterogeneous branches combined with a dynamic aggregation strategy effectively facilitate the extraction and interaction of low-level modality-specific features. The subsequent LCAA and RepFFN modules handle contextual refinement and nonlinear enhancement, respectively, forming a cohesively designed, lightweight fusion module together with RepMBConv.

Training Overhead Clarification: The reported training memory usage increases from 2676 MB to 3946 MB, which alleviates my concern that the multi branch channel expansion in RepMBConv would introduce prohibitive overhead.

Architectural Clarification: The authors explicitly acknowledged that the omission of the max pooling branch in Table 7 was an oversight and stated that it will be corrected in the revised manuscript. I appreciate that this issue was addressed directly.

In summary, MobileFusion presents a novel reparameterization-based architecture tailored to multi-modal fusion. It achieves strong fusion performance while maintaining an extremely low parameter count and low inference cost, and the additional robustness experiments further support its practical value. Overall, the work is a technically sound and practically relevant contribution toward high-quality, efficient image fusion on resource-constrained edge devices, and I am therefore raising my score.

**Key Questions For Authors:**

1. The input image has already undergone channel stitching before entering RepMBConv. Please explain how the subsequent multi-branch structure achieves its claimed "active" cross-modal interaction, rather than the extraction of pre-mixed features.
2. Ablation experiments show that "stitching and reprojection" is superior to element-level addition, but will its high-dimensional channel projection during training significantly increase memory pressure?
3. Section 3.3 and Figure 2(b) describe in detail a two-stream attention mechanism that utilizes both average pooling and max pooling, but the detailed architecture in Table 7 does not include a max pooling branch. Could the authors clarify which architecture is actually being evaluated?

**Limitations:**

See the weaknesses section.

**Strengths And Weaknesses:**

Strengths:
1. This paper achieves real-time infrared and visible light image fusion on resource-constrained mobile devices.
2. The RepMBConv module undergoes structural reparameterization to facilitate cross-modal interaction and simplifies it into an ultra-high-speed single-path pipeline during inference.
3. Extensive experiments demonstrate that this method achieves an excellent performance-to-priority balance, enabling MobileFusion to achieve highly competitive visual quality with only approximately 4K parameters, while maintaining real-time execution speeds on edge platforms such as Jetson Orin NX.

Weaknesses:
1. Limited Technological Innovation: The proposed method is primarily based on structure reparameterization and a two-stream attention mechanism. Its main components, including RepVGG-style modules and standard channel/spatial attention mechanisms, are simple assembly of existing mature technologies.
2. Limited Methodological Contribution: Although the authors claim their design is "fusion-oriented," modifications such as replacing element-wise summation with cascading in the RepMBConv module do not represent theoretical breakthroughs or the introduction of new operators.

---

> ### Author Rebuttal · Authors · 2026-03-31
>
> We sincerely appreciate the reviewer’s positive feedback.
>
> Overall response: Our goal is not to introduce a fundamentally new primitive operator or a formal theoretical breakthrough. Instead, our contribution is a fusion-oriented and deployment-aware architecture design for mobile infrared-visible image fusion under strict efficiency constraints. RepMBConv is not a direct reuse of RepVGG-style design, as it explicitly introduces heterogeneous branches and dynamic aggregation to selectively preserve critical complementary cross-modal cues, which conventional Rep designs do not address in low-level fusion. Moreover, LCAA and RepFFN are not independent components. Together with RepMBConv, they form a unified lightweight block in which interaction modeling, context refinement, and nonlinear enhancement are co-designed, and the ablation studies verify that the performance gains come from this synergy rather than from a simple assembly of mature techniques. With only around 4K parameters, MobileFusion achieves the fastest inference speed on mobile platforms while delivering the best overall fusion performance, significantly improving the practical applicability of IVIF in resource-constrained mobile scenarios.
>
> Q1: Cross-modal interaction in our method is driven by the fusion-oriented structural design of RepMBConv. The initial channel stitching only provides the network with raw bimodal input and does not by itself establish meaningful interaction. At this stage, the representation remains essentially pre-mixed. The actual interaction is induced within the multi-branch architecture. Specifically, the key to RepMBConv lies in its heterogeneous multi-branch design, in which convolutional branches with different receptive fields and directional sensitivities process the concatenated input in parallel and independently. Each branch produces a distinct intermediate feature response, which encourages the network to explore rich complementary information between the infrared and visible modalities across multiple scales and orientations(e.g., edge alignment, thermal contrast, structural consistency). The subsequent concatenation-projection layer learns a set of linear combination weights to adaptively and effectively integrate the complementary cues extracted by different branches, acting as a lightweight yet powerful cross-modal aggregation mechanism. Therefore, RepMBConv does not merely operate on pre-mixed features. Instead, it explicitly establishes and strengthens an inductive bias for cross-modal interaction during training.
>
> Q2: Although concatenation followed by channel projection introduces additional parameters compared to element-wise addition, it incurs only a modest increase in GPU memory consumption under the same training configuration (batch size = 12), rising from 2,676 MB to 3,946 MB. This memory usage is well within the capacity of an NVIDIA GeForce RTX 4090 and remained manageable under our training setup. Importantly, it is confined to the training stage. During inference, our architecture is reparameterized into a structurally compact form, eliminating the overhead of the multi-branch design and resulting in a highly efficient deployment model.
>
> Q3: The dual-stream attention mechanism employs both adaptive average pooling and max pooling to extract global statistical information, where the max-pooling branch is introduced to better capture salient responses in infrared images. This is consistent with the description in Section 3.3 and Figure 2(b). Table 7 indeed omits the max-pooling branch, and we will correct this in the revised manuscript. We sincerely thank the reviewer for pointing this out.

---

> > ### Author Rebuttal · Reviewer_z81f · 2026-04-04
> >
> > The authors clearly addressed my all concerns, which strengthens the paper’s technical clarity and overall contribution.
> >
> > Clarification on Cross-Modal Interaction: The authors clearly explain how RepMBConv differs from conventional Rep designs: its channel-expanded heterogeneous branches combined with a dynamic aggregation strategy effectively facilitate the extraction and interaction of low-level modality-specific features. The subsequent LCAA and RepFFN modules handle contextual refinement and nonlinear enhancement, respectively, forming a cohesively designed, lightweight fusion module together with RepMBConv.
> >
> > Training Overhead Clarification: The reported training memory usage increases from 2676 MB to 3946 MB, which alleviates my concern that the multi branch channel expansion in RepMBConv would introduce prohibitive overhead.
> >
> > Architectural Clarification: The authors explicitly acknowledged that the omission of the max pooling branch in Table 7 was an oversight and stated that it will be corrected in the revised manuscript. I appreciate that this issue was addressed directly.
> >
> > In summary, MobileFusion presents a novel reparameterization-based architecture tailored to multi-modal fusion. It achieves strong fusion performance while maintaining an extremely low parameter count and low inference cost, and the additional robustness experiments further support its practical value. Overall, the work is a technically sound and practically relevant contribution toward high-quality, efficient image fusion on resource-constrained edge devices, and I am therefore raising my score to 5.

---

> > > ### Author Response · Authors · 2026-04-05
> > >
> > > We are deeply grateful for your insightful comments and positive feedback. Thank you for your support and endorsement of our work.

---

### Decision · Program_Chairs · 2026-04-30

**Decision:**

Accept (regular)

**Comment:**

This paper proposes MobileFusion, a lightweight convolutional framework for infrared-visible image fusion (IVIF) designed for real-time deployment on resource-constrained devices. The method is built upon a re-parameterizable multi-branch convolution module, which enables rich cross-modal feature interaction during training while collapsing into a single-path operator at inference time for efficiency.

This paper presents a well-engineered and meaningful system for efficient multi-modal image fusion. Its primary contribution lies in achieving an excellent quality–efficiency trade-off, which is demonstrated and strengthened after rebuttal. However, several concerns remain: (i) While the empirical validation is improved, it is not fully comprehensive. (ii) The work provides limited conceptual or theoretical insight into multi-modal fusion. Despite these limitations, the paper makes a clear contribution in the direction of deployment-oriented image fusion. Given the strong practical value and solid experimental support, the AC recommends acceptance.